# Striatal direct and indirect pathway neurons differentially control the encoding and updating of goal-directed learning

James Peak, Billy Chieng, Genevra Hart[†], Bernard W Balleine[†]*

Decision Neuroscience Lab, School of Psychology, UNSW Sydney, Sydney, Australia

**Abstract** The posterior dorsomedial striatum (pDMS) is necessary for goal-directed action; however, the role of the direct (dSPN) and indirect (iSPN) spiny projection neurons in the pDMS in such actions remains unclear. In this series of experiments, we examined the role of pDMS SPNs in goal-directed action in rats and found that whereas dSPNs were critical for goal-directed learning and for energizing the learned response, iSPNs were involved in updating that learning to support response flexibility. Instrumental training elevated expression of the plasticity marker Zif268 in dSPNs only, and chemogenetic suppression of dSPN activity during training prevented goal-directed learning. Unilateral optogenetic inhibition of dSPNs induced an ipsilateral response bias in goal-directed action performance. In contrast, although initial goal-directed learning was unaffected by iSPN manipulations, optogenetic inhibition of iSPNs, but not dSPNs, impaired the updating of this learning and attenuated response flexibility after changes in the action-outcome contingency.

**\*For correspondence:**
bernard.balleine@unsw.edu.au

[†]These authors contributed equally to this work

**Competing interests:** The authors declare that no competing interests exist.

## Introduction

Animals adapt to changing environments and maximize opportunities for reward by flexibly adjusting their actions according to current goals. Such goal-directed actions rely on the ability of animals to encode the relationship between an action and its consequences or outcome, and to integrate that information with the current value of the outcome (*Balleine, 2019*; *Balleine and Dickinson, 1998*). In the brain, the acquisition of goal-directed actions is mediated by a corticostriatal circuit centred on the posterior dorsomedial striatum (pDMS), which is thought to influence the performance of, and choice between, goal-directed actions via local changes in its output to the substantia nigra pars reticulata (SNr; *Balleine, 2019*; *Peak et al., 2019*, for reviews). Nevertheless, although the function of these output pathways has been a major focus of recent research, the precise nature of the changes in the pDMS necessary for the acquisition and performance of goal-directed action remains unresolved.

Analyses of striatal output have long focused on the influence of two pathways, one projecting directly to the SNr and the other indirectly via the external segment of the globus pallidus (GPe) and subthalamic nucleus (*Albin et al., 1989*). Canonically, each pathway is thought to be controlled by a distinct population of spiny projection neuron (SPN); the direct pathway SPNs (dSPNs) that predominantly express the dopamine D1 receptor (D1R), and the indirect pathway SPNs (iSPNs) that express the dopamine D2 receptor (*Gerfen and Surmeier, 2011*; *Calabresi et al., 2014* for reviews). To date, however, studies investigating these output pathways have been almost entirely concerned with their modulation of spontaneous or previously acquired responses. These studies have found that bilateral manipulation of dSPNs versus iSPNs in the pDMS produces differential effects on spontaneous movement or on the flexible performance of reward-related actions (*Kravitz et al., 2010*;

*Kravitz et al., 2012*; *Nonomura et al., 2018*; *Hikida et al., 2010*; *Kwak and Jung, 2019*), whereas unilateral dSPN and iSPN stimulation alters movements and action selection in contralateral and ipsilateral action space, respectively (*Kravitz et al., 2010*; *Bay König et al., 2019*; *Tai et al., 2012*). Nevertheless, no studies to date have sought to establish whether these effects are specific to goal-directed action control. Furthermore, despite a number of studies implying that dSPNs play a role in goal-directed learning, reporting increases in learning-induced neuronal activity (*Maroteaux et al., 2014*; *Matamales et al., 2020*; *Shan et al., 2014*) and synaptic plasticity (*Shan et al., 2014*), it remains unclear whether one or other or, indeed, both of these pathways are necessary for the acquisition of goal-directed action.

The current series of experiments sought, therefore, to evaluate and compare the role of striatal output pathways in the acquisition and flexible performance of goal-directed action. Importantly, prior studies have universally employed cell-type specific manipulations; typically using BAC transgenic mice to individually manipulate D1- and D2- (or A2A) expressing SPNs as proxies for the output pathways themselves, which has the advantage of manipulating a large proportion of each cell type but has the disadvantage of doing so indiscriminately. This may be important; considerable evidence suggests that D1- and D2-expressing SPNs can interact locally (*Taverna et al., 2008*; *Matamales et al., 2020*), and at their downstream target in the GPe (*Cazorla et al., 2014*; *Wu et al., 2000*; *Fujiyama et al., 2011*; *Kawaguchi et al., 1990*; see *Burke et al., 2017* for review). As a consequence, we chose instead to target the specific output pathways from the pDMS based on their projections. To achieve this, we employed a pathway-specific approach using the retrograde transport of Cre to differentially label striatal neurons projecting directly to the SNr versus the GPe using DREADD and optogenetic tools to manipulate Cre-expressing neurons. Using this approach, we found clear evidence that, within the pDMS, dSPNs but not iSPNs are necessary to encode the action-outcome associations necessary to acquire goal-directed actions, whereas iSPNs but not dSPNs are necessary flexibly to adjust performance when the action-outcome contingencies controlling goal-directed action change.

## Results

### The acquisition of goal-directed action induces Zif268 expression in dSPNs in the pDMS

We first sought to establish whether plasticity-related activity differs in dSPNs and iSPNs during the acquisition of goal-directed action. Based on the above considerations, we used a pathway-specific approach, targeting dSPNs and iSPNs by injecting the retrograde tracers, fluorogold (FG) and cholera-toxin B (CTB) subunit, bilaterally into the direct monosynaptic targets of each pathway, the SNr and GPe, respectively (*Figure 1A and B*). The spread of these injection sites is presented in *Figure 1—figure supplement 1A*. We were able to identify dSPNs and iSPNs in the same animal; *Figure 1C* (also see *Figure 1—figure supplement 1B*) shows triple labelling of SPNs with either FG or CTB or both, co-labelled with DARPP-32, a marker for SPNs. There was no difference in the number of cells/mm$^2$ labelled with FG or CTB (*Figure 1—figure supplement 1C*; $F(1,20)=1.26$, $p=0.276$) or in the mean percentage of neurons *co*-labelled with either FG or CTB (*Figure 1D*; $F(1,20)=1.2$, $p=0.279$), suggesting that this approach was not significantly biased towards one or other population of SPNs. Overall, we were able to target approximately half of all SPNs identified with DARPP-32. There was a substantial population of cells labelled with both tracers, suggesting a degree of overlap in collaterals projecting from both cell types to the SNr and GPe (estimated at 16–17% of the total number of retrogradely labelled cells).

Next, we assessed the learning-related activity in dSPNs and iSPNs induced by instrumental training. Half of the rats were given training over 4 days, in which pressing a lever (located to the left or right side of a central food magazine) delivered grain pellets on increasing random interval schedules of reinforcement (see Materials and methods). A second, control group, received yoked training in which grain pellets were delivered at the same average interval as the instrumentally trained rats, but independently of lever pressing. This resulted in robust acquisition of the instrumental response for rats in Group Instrumental, but not for rats in Group Yoked (*Figure 1E*; $F(1,20)=90.73$, $p<0.001$). Both groups entered the magazine to retrieve the food, and although Group Instrumental had numerically higher entry rates, this difference was not significant (*Figure 1F*; $F(1,20)=4.04$, $p=0.058$).

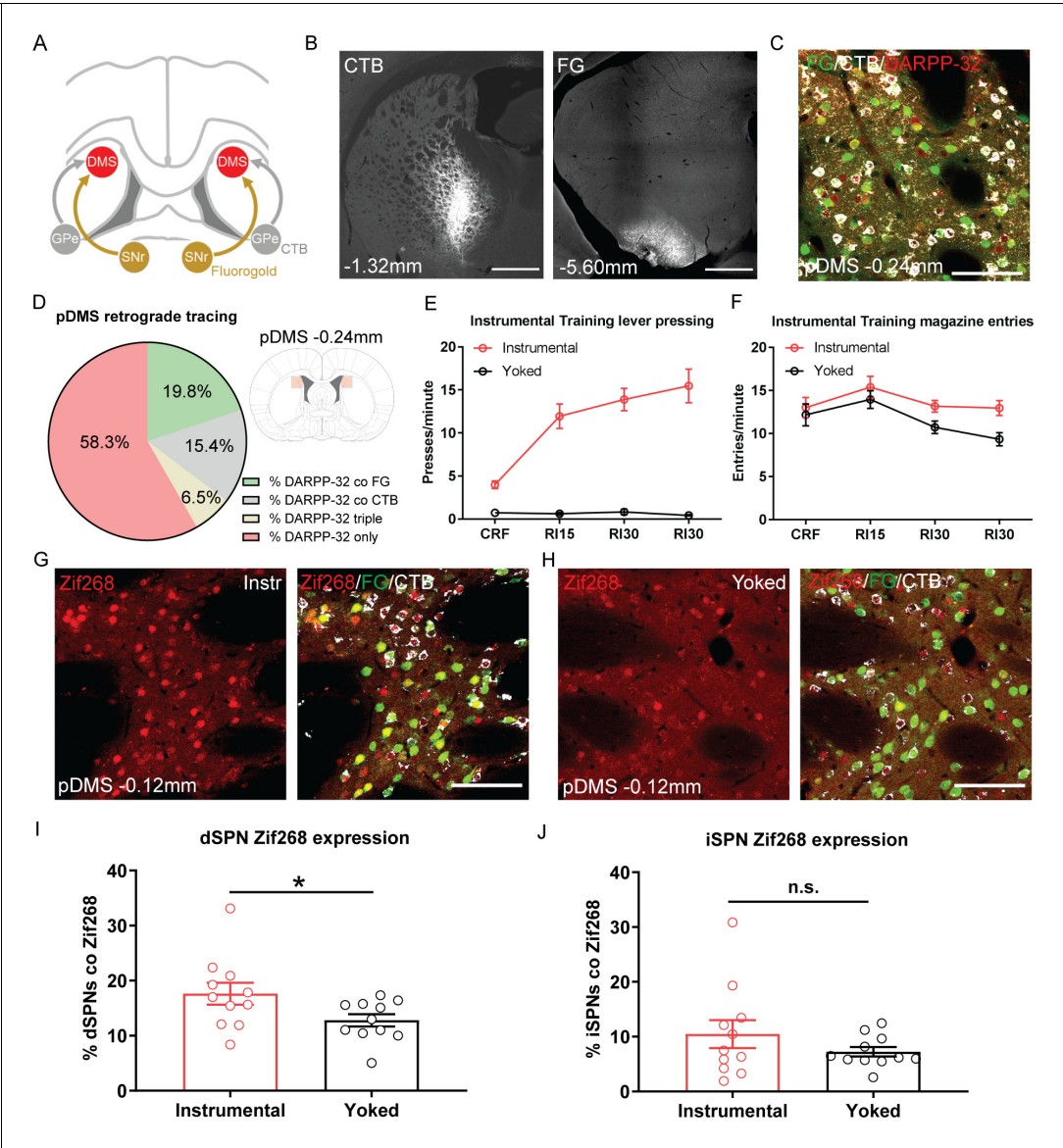

**Figure 1.** The acquisition of goal-directed actions induces response-specific expression of Zif268 in direct spiny projection neurons (dSPNs) in the posterior dorsomedial striatum (pDMS). (**A**) Schematic illustrating the surgery design for retrograde tracing. Rats received bilateral infusions of the retrograde tracers fluorogold (FG) and cholera-toxin B (CTB) into the substantia nigra pars reticulata (SNr) and globus pallidus (GPe) respectively. (**B**) Fluorescent confocal images showing injection sites of CTB (left) and FG (right) in the GPe and SNr respectively; scale bars, 1000 μm. (**C**) Fluorescent confocal image taken from one 40 μm coronal section in the pDMS, illustrating fluorescence labelling of DARPP-32 (red), FG (green), and CTB (white); scale bar, 100 μm. (**D**) Percentages represent the proportion of total DARPP-32 positive SPNs that also express FG or CTB in the pDMS (imaging region indicated in pale red, right). Data presented are mean percentages for 22 rats, averaged across hemispheres. (**E**) Mean (± SEM) lever presses per minute averaged across each day of instrumental or yoked training for each group. (**F**) Mean (± SEM) magazine entries per minute averaged across each day of instrumental or yoked training for each group. (**G**) Fluorescent confocal image of a coronal 40 μm section for an instrumentally trained rat; left image shows Zif268 expression (red), and right image shows the same merged with FG labelled dSPNs (green) and CTB labelled indirect spiny projection neurons (iSPNs; white); scale bar, 100 μm. (**H**) Same as G but for a yoked trained rat. (**I**) Percentage of labelled dSPNs that were co-labelled with Zif268 for each rat in Group Instrumental and Group Yoked. Data presented are means of four pDMS sections per rat. Bars represent group means ± SEM. (**J**) Same as I but for iSPNs. *p<0.05.

The online version of this article includes the following figure supplement(s) for figure 1:

**Figure supplement 1.** Projection target verification for the SPN tracing and immunofluorescence experiment.

Ninety minutes after the beginning of the final training session, rats were perfused and we used immunofluorescence to measure the expression of the immediate-early gene Zif268 (also known as EGR1), widely used as an activity marker for learning-related plasticity in the striatum (*Moratalla et al., 1992*; *Knapska and Kaczmarek, 2004*; *Hernandez et al., 2006*; *Maroteaux et al., 2014*), in dSPNs and iSPNs in the pDMS. There was robust labelling of Zif268 in both groups (*Figure 1G and H*). *Figure 1I and J* shows the percentage of dSPNs and iSPNs co-labelled with Zif268, respectively. Rats in Group Instrumental had significantly higher Zif268 expression in dSPNs than rats in Group Yoked ($F_{(1,20)}=4.52$, $p=0.046$). In contrast, analysis of Zif268 expression in iSPNs revealed no significant differences between Group Instrumental and Group Yoked in overall levels of Zif268 expression ($F_{(1,20)}=1.43$, $p=0.245$).

## Goal-directed learning is disrupted by chemogenetic inhibition of dSPNs but not iSPNs in the pDMS

We next sought to assess the causal involvement of dSPNs and iSPNs in goal-directed learning. To induce prolonged suppression of neuronal activity across each day of instrumental training, chemogenetic (DREADDs) inhibition is optimal; however, the use of DREADDs in striatal neurons has been a topic of debate in the literature with concerns raised over its effectiveness. Although there have been successful reports of the use of DREADDs in striatal neurons (*Ferguson et al., 2011*; *Ferguson et al., 2013*; *Carvalho Poyraz et al., 2016*), SPNs have a relatively low number of G-protein-activated inwardly rectifying potassium channels (*Lovinger, 2010*). We therefore assessed whether hM4Di DREADDs expressed on SPNs could suppress cortically evoked neuronal firing following the application of the synthetic ligand of the hM4Di receptor, clozapine-N-oxide (CNO; RTI International).

We used a dual-virus approach (*Figure 2A*; *Marchant et al., 2016*; *Hart et al., 2018a*) to express hM4Di receptors on dSPNs or iSPNs, by bilaterally infusing a retrogradely transported AAV-virus expressing Cre recombinase (rAAV5.CMV.HI.eGFP-CRE.WPRE.SV40; retro-Cre) into either the SNr or GPe to express Cre on dSPNs or iSPNs, respectively. We then injected a Cre-dependent hM4Di DREADDs virus (rAAV5/hSyn-DIO-hM4D(Gi)-mCherry; DIO-hM4D) or a control fluorophore lacking the hM4D construct (rAAV5/hSyn-DIO-mCherry; DIO-mCherry) into the pDMS, to selectively express DIO-hM4D or DIO-mCherry on either dSPNs or iSPNs. To induce neuronal firing in SPNs, we injected channelrhodopsin (AAV5-CaMKIIa-hChR2(H134R)-eYFP; ChR2-eYFP) into the prelimbic cortex (*Figure 2A*), which has dense bilateral glutamatergic projections onto dSPNs and iSPNs (*Hart et al., 2018a*; *Wall et al., 2013*). We used ex vivo slice electrophysiology to assess the effect of CNO on the cortically evoked firing of dSPNs and iSPNs expressing hM4Di DREADDs.

*Figure 2B* shows the raw trace recorded from a dSPN (top) and an iSPN (bottom) expressing DIO-hM4D; action potentials were elicited by pulsing an LED light onto ChR2-containing corticostriatal axons (blue bars). Group CNO control consisted of cells expressing DIO-mCherry on dSPNs as well as iSPNs. A raw trace recorded from a dSPN expressing DIO-mCherry and confocal images of one representative recorded neuron from each group are presented in *Figure 2—figure supplement 1A and B*. We compared action potential frequency at three time points; 1 min prior, immediately prior, and 1 min after CNO application. There was no difference in any group in action potential frequency between the two pre-CNO time points (*Figure 2C*; pairwise comparisons, highest $F_{(1,21)}=2.68$, $p=0.117$, iSPN hM4D). Following the administration of CNO, action potential frequency was significantly reduced relative to the time point immediately prior to CNO application in Group dSPN hM4D and iSPN hM4D but not in Group CNO control (pairwise comparisons, Group CNO control, $F_{(1,21)}=0.93$, $p=0.346$; Group dSPN hM4D, $F_{(1,21)}=8.96$, $p=0.007$; Group iSPN hM4D, $F_{(1,21)}=15.8$, $p=0.001$).

Having established that hM4Di DREADDs expressed on SPNs could suppress cortically evoked firing, we next assessed whether this activity was necessary for goal-directed learning. We used the same dual-virus approach to express either DIO-hM4D or DIO-mCherry on dSPNs (*Figure 2D*) or iSPNs (*Figure 2F*). Example images of retro-Cre expression in the SNr (left) and DIO-hM4D expression on pDMS dSPNs (right) are presented in *Figure 2E*. The same is shown for retro-Cre expression in the GPe and DIO-hM4D expression on pDMS iSPNs in *Figure 2G*. The spread of retro-Cre and DIO-hM4D expression in the SNr or GPe and pDMS respectively in all included animals is presented in *Figure 2—figure supplement 1C–E*.

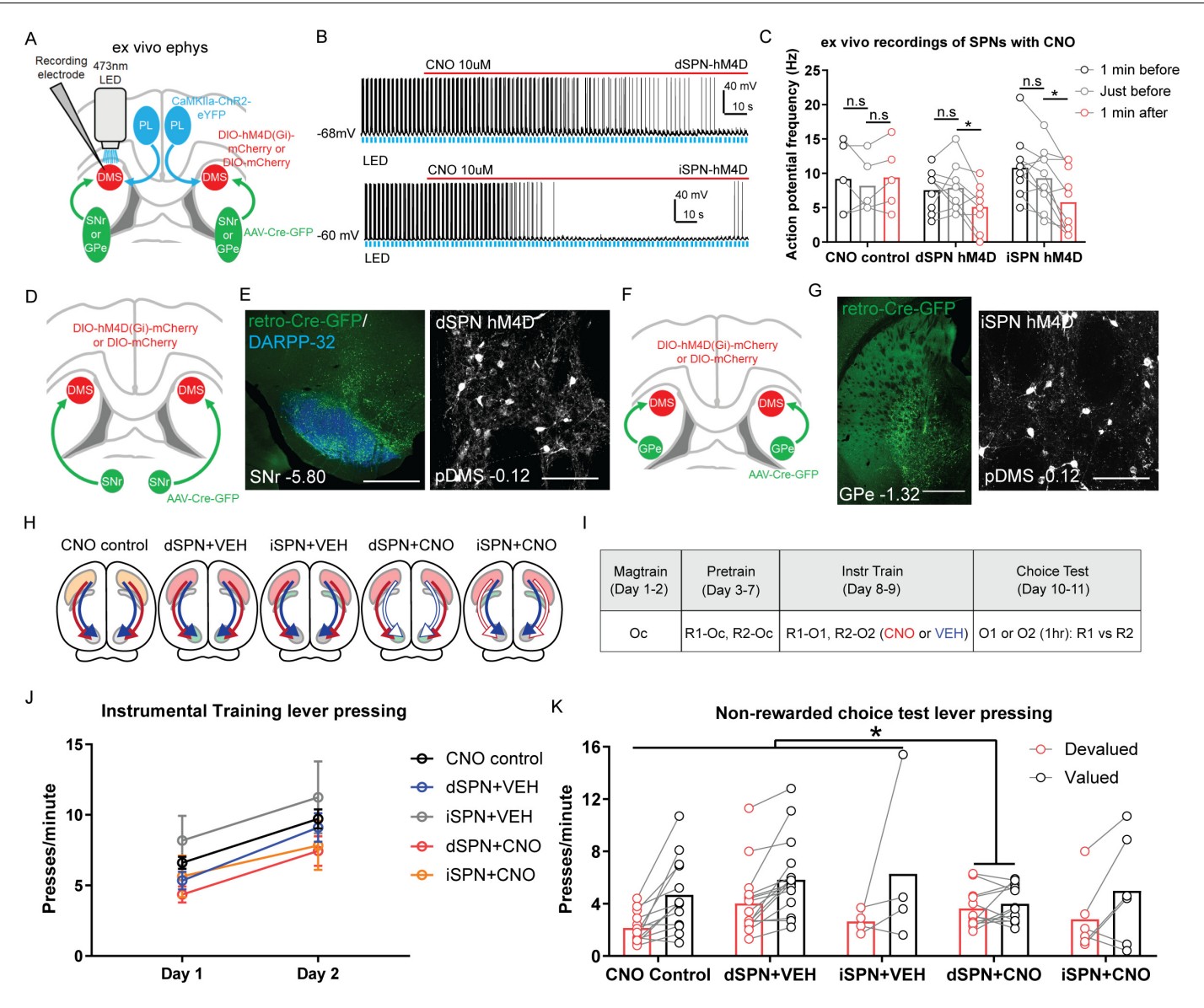

**Figure 2.** Goal-directed learning is disrupted by chemogenetic inhibition of direct spiny projection neurons (dSPNs) but not indirect spiny projection neurons (iSPNs) in the posterior dorsomedial striatum (pDMS). (**A**) Schematic depicting the viral injections for ex vivo slice electrophysiology recordings; retro-Cre was injected into the substantia nigra pars reticulata (SNr) or globus pallidus (GPe), DIO-hM4D or DIO-mCherry was injected into the pDMS and ChR2-eYFP injected into the PL; direction of arrows indicates retrograde or anterograde transport of the virus. (**B**) Example trace from one recorded dSPN (top) and one recorded iSPN (bottom) expressing DIO-hM4D. Upward deflection black lines represent action potentials, which were elicited by the pulsing of a 473 nm wavelength LED light (blue) onto corticostriatal terminals (see Materials and methods). The red line indicates the time period when 10 µM clozapine-N-oxide (CNO) solution was added to the extracellular solution. (**C**) Action potential frequency of dSPNs and iSPNs elicited by light-evoked terminal glutamate release from ChR2-containing corticostriatal axons in the pDMS 1 min before (black), immediately before (grey) and 1 min after (red) the application of CNO in the extracellular solution. Each data point represents individually recorded cells. (**D**) Schematic depicting the viral injections to target dSPNs; retro-Cre was injected bilaterally into the SNr and DIO-hM4D or DIO-mCherry was injected bilaterally into the pDMS. (**E**) Confocal image (scale bar, 1000 µm) from one rat showing retro-Cre expression in the SNr (left) and a representative confocal image (scale bar, 100 µm) showing DIO-hM4D expression in pDMS dSPNs (right). (**F**) Same as D but for targeting iSPNs; retro-Cre was injected into the GPe. (**G**) Same as E but showing retro-Cre expression in the GPe and DIO-hM4D expression in pDMS iSPNs. (**H**) Summary of experimental groups; blue arrows indicate intact direct pathway function, red arrows indicate intact indirect pathway function, and unfilled arrows indicate inhibited pathway. (**I**) Summary of the experimental design; R1 and R2 indicate left and right lever responses; Oc, O1, and O2 indicate distinct food outcomes; CNO and VEH indicate injections of clozapine-N-oxide or vehicle, respectively. (**J**) Mean (± SEM) lever presses per minute averaged across each day of instrumental training for each group. (**K**) Mean lever presses per minute on the devalued and valued lever for each rat in each group, averaged across two choice tests under extinction. For all data, bars represent group means. *p<0.05.

*Figure 2 continued on next page*

*Figure 2 continued*

The online version of this article includes the following figure supplement(s) for figure 2:

**Figure supplement 1.** Verification of hM4D virus and supplemental behavior for the DREADDs suppression experiment.

We quantified the mean number of cells expressing DIO-hM4D or DIO-mCherry for each animal (*Figure 2—figure supplement 1F*); there was no difference in the number of dSPNs or iSPNs expressing DIO-hM4D (dSPN+VEH versus dSPN+CNO, F(1,46)=0.06, p=0.808; iSPN+VEH versus iSPN+CNO, F(1,46)=0.52, p=0.473). There were, however, significantly more SPNs (half dSPNs and half iSPNs) expressing DIO-mCherry (Group CNO control versus rest; F(1,46)=23.93, p<0.001). We compared the rate of infection achieved with this dual-virus approach in dSPNs to that obtained using retrograde tracing (see *Figure 1—figure supplement 1C*); retro-Cre in combination with DIO-hM4D achieved 38–42% of the labelling achieved with FG, and this percentage is consistent with that previously reported using the same viral combination in the PL-pDMS pathway, and compared to FG (e.g., 30–45% estimated by *Hart et al., 2018a*). For iSPNs, retro-Cre in combination with DIO-hM4D achieved 81–102% of the labelling that was achieved with CTB. The substantially higher infection rate from GPe is likely to be due to a combination of two things; fewer CTB labelled neurons compared to FG (see *Figure 1—figure supplement 1C*) and more efficient retro-Cre transport in striatopallidal neurons, possibly due to the shorter pathway length.

For behavioural testing there were five groups of rats, summarized in *Figure 2H*. There were three control groups: (1) Group CNO control, expressed DIO-mCherry on SPNs (half on dSPNs and half on iSPNs which were pooled based on a lack of significant differences in press rate during training or test; Fs <3.69, lowest p=0.081); (2) Group dSPN+VEH, and (3) Group iSPN+VEH both of which expressed DIO-hM4D on dSPNs or iSPNs, respectively, and received injections of vehicle throughout training. The two experimental groups expressed DIO-hM4D and received injections of CNO throughout training to suppress the activity of dSPNs (Group dSPN+CNO) or iSPNs (Group iSPN+CNO).

A summary of the behavioural design is presented in *Figure 2I*. Rats were given 5 days of instrumental pre-training with two levers for a common outcome (Oc) after which they received 2 days of instrumental training with the same two levers paired with new, distinct outcomes (R1–O1 and R2–O2). Intraperitoneal injections of CNO or VEH were administered 1 hr prior to these training sessions. All rats acquired the instrumental response across pre-training (*Figure 2—figure supplement 1G*). Across 2 days of instrumental training (*Figure 2J*), all rats increased their response rates from Day 1 to Day 2 (F(1,46)=83.28, p<0.001) and there were no significant differences between groups in overall response rates (Bonferroni, k=3, highest F(1,46)=4.86, p=0.099, dSPN+CNO versus controls).

In order to assess goal-directed learning, we gave rats a choice extinction test, drug free, immediately after specific satiety-induced outcome devaluation. Prior to the test rats were allowed freely to consume one of the previously earned outcomes (O1 or O2) for 1 hr after which they were returned to the operant chambers and tested with both levers available in a choice extinction test. This test was conducted twice so that each outcome was devalued across tests and each lever served as the devalued lever, allowing the effect of devaluation to be assessed within subjects. Goal-directed action is demonstrated by rats showing a reduction in responding on the lever previously associated with the now-devalued outcome relative to the other lever, indicating their ability to integrate prior action-outcome learning with the current value of the outcome. Responding on the valued and devalued levers is presented for each rat in each group in *Figure 2K*. We compared responding in Groups dSPN+CNO or iSPN+CNO against the three control groups; dSPN+VEH, iSPN+VEH, and CNO control. We also compared the control groups that received vehicle to the CNO control. There were no differences between groups in overall rates of responding (main effect of group, highest F(1,46)=2.01, p=0.163, VEH controls versus CNO control). There was a main effect of devaluation (Bonferroni, k=3, F(1,46)=28.73, p<0.001) and, more importantly, a significant interaction between group (dSPN+CNO versus controls) and devaluation (Bonferroni, k=3, F(1,46)=6.93, p=0.012) indicating that whereas the control groups showed a reliable devaluation effect, this was abolished in rats that had dSPNs suppressed during instrumental training. Importantly, there was no difference in the magnitude of the devaluation effect in Group iSPN+CNO relative to the three control groups, or

between the two vehicle groups and Group CNO control (Fs <1, lowest p=0.677). These results suggest that dSPN activity in the pDMS is necessary during training for animals to encode the specific action-outcome associations necessary to acquire goal-directed instrumental actions.

In a subset of rats, we examined whether this effect was attributable to a diminished capacity to discriminate between responses or outcomes, when dSPNs or iSPNs were suppressed. To achieve this, rats were injected with CNO or vehicle (group allocations were unchanged) and given a rewarded choice test following outcome devaluation (*Figure 2—figure supplement 1H*), in which each lever delivered its respective outcome as in training, negating the need for animals to use prior action-outcome learning to bias actions. Under these circumstances, all rats were able to respond according to current outcome value, and there were no differences between groups (main effect of lever, Bonferroni, k=3, F(1,18)=53.24, p<0.001; no effect of group and no group × lever interactions, Fs <1.6, lowest p=0.224). This result demonstrates that, under dSPN (or iSPN) inhibition, rats can discriminate the lever press actions and outcomes and that specific satiety-induced outcome devaluation is effective in biasing choice when animals are given feedback on those choices, confirming that the effects of dSPN inactivation were specific to goal-directed learning.

Finally, although we saw no evidence for motor disruption across training or during the rewarded choice test when activity in dSPNs or iSPNs was suppressed, we also assessed whether suppressing activity in pDMS SPNs induces a locomotor impairment more directly using a rotarod test (*Hamm et al., 1994*). Rats infused with DIO-hM4D were all tested twice; once under vehicle and once under CNO. As shown in *Figure 2—figure supplement 1I*, suppressing SPNs did not affect the amount of time spent on the rotarod for rats in any group (pairwise comparisons between CNO and VEH in each group, highest F=2.7, dSPN+VEH). Therefore, although it remains possible that fine motor movements were affected by inactivation, the suppression of dSPNs or iSPNs produced no overt impairments detectable using this test.

## Chemogenetic stimulation of iSPNs in the pDMS leaves goal-directed learning intact

The results so far suggest that goal-directed learning relies on dSPN plasticity during training; however, it is possible that learning was disrupted because dSPN inhibition altered the relative output of dSPNs and iSPNs to favour heightened iSPN output. To assess this possibility, we tested whether increased iSPN activity could similarly affect goal-directed learning. We used a Cre-dependent hM3Dq DREADD (rAAV5/hSyn-DIO-hM3D-mCherry; DIO-hM3D) to stimulate the activity of iSPNs targeted using the same retro-Cre approach described (*Figure 3A and B*). We used slice electrophysiology to verify that CNO increased neuronal firing in iSPNs expressing Cre-dependent hM3Dq DREADDs; application of CNO produced a significant increase in the number of action potentials recorded from iSPNs expressing DIO-hM3D, relative to baseline (*Figure 3D and E*; paired t-test, p=0.049). A confocal image of a representative recorded neuron is presented in *Figure 3—figure supplement 1A*.

Next, we assessed the functional consequences of iSPN stimulation on goal-directed learning using the behavioural design described previously (*Figure 2I*). Expression of retro-Cre in the GPe and DIO-hM3D in the pDMS is shown in *Figure 3B* and the location and spread of virus expression for all included animals are shown in *Figure 3—figure supplement 1B and C*. The mean number of cells expressing DIO-mCherry or DIO-hM3D was quantified and is presented in *Figure 3—figure supplement 1D*; there were no significant differences between groups or viruses (highest F(1,17) =2.51, p=0.131). There were three groups of rats, summarized in *Figure 3C*: all groups received retro-Cre into the GPe. There were two control groups; one expressed DIO-mCherry on iSPNs and received CNO injections throughout training and the other expressed DIO-hM3D on iSPNs and received VEH injections throughout training. The third group expressed DIO-hM3D on iSPNs and received CNO injections throughout training to stimulate the activity of iSPNs in the pDMS. We found no effect of stimulating the activity of iSPNs during instrumental training on press rates (*Figure 3F*; Fs <1, lowest p=0.391), and found that this stimulation did not affect goal-directed learning (*Figure 3G*; main effect of devaluation, F(1,17)=34.561, p<0.001; no devaluation × group interaction, Fs <1.1, lowest p=0.313). Therefore, neither inhibition nor stimulation of iSPNs during instrumental training produced any detectable effect on goal-directed learning.

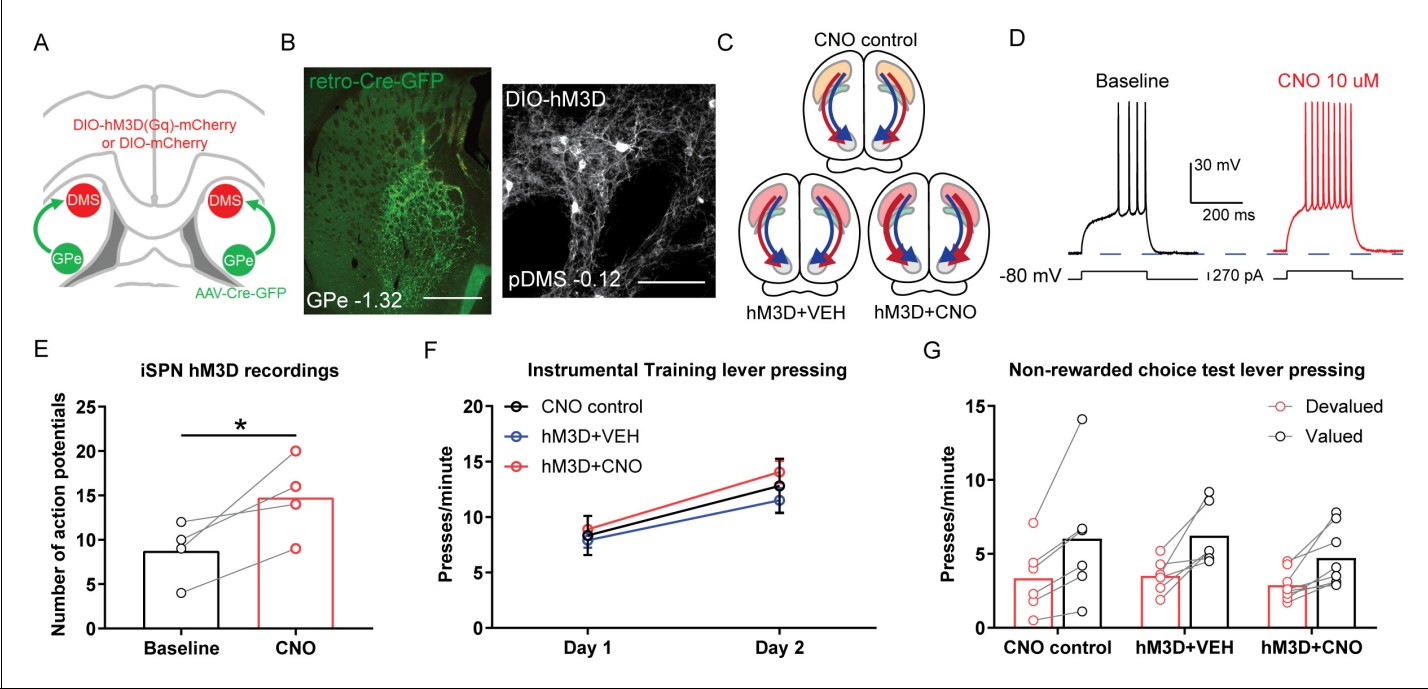

**Figure 3.** Chemogenetic stimulation of indirect spiny projection neurons (iSPNs) in the posterior dorsomedial striatum (pDMS) leaves goal-directed learning intact. (**A**) Schematic depicting the viral injections, retro-Cre was injected bilaterally into the globus pallidus (GPe) and DIO-hM3D or DIO-mCherry was injected bilaterally into the pDMS. (**B**) Confocal image (scale bar, 1000 μm) from one rat showing retro-Cre expression in the GPe (left) and a confocal image (scale bar, 100 μm) from one rat showing DIO-hM3D expression on pDMS iSPNs. (**C**) Summary of experimental groups; blue arrows represent direct pathway, red arrows represent indirect pathway, and thicker arrows indicate increased activity in stimulated pathway. (**D**) An example trace from one iSPN from injection of a depolarizing current step (at resting membrane potential) before and after bath application of clozapine-N-oxide (CNO). (**E**) Number of action potentials evoked from an identical size current step injection before and during CNO application in hM3D-expressing iSPNs. The chosen step size in each neuron was when current injection first elicited action potentials, before drug. Individual data points represent each recorded cell. (**F**) Mean (± SEM) lever presses per minute averaged across each day of instrumental training for each group. (**G**) Mean lever presses per minute on the devalued and valued lever for each rat in each group, averaged across 2 days of non-rewarded choice tests. For all data, bars represent group means. *p<0.05.
The online version of this article includes the following figure supplement(s) for figure 3:

**Figure supplement 1.** Verification of the hM3D virus for the DREADDs stimulation experiment.

## Bilateral optogenetic inhibition of dSPNs or iSPNs does not affect the performance of goal-directed actions

Having confirmed that goal-directed learning is dependent on the activity of dSPNs but not iSPNs, we assessed the involvement of these pathways in goal-directed performance. Here we employed an optogenetic approach for temporally precise inhibition of projection neurons during action performance. We used the same dual-virus approach described above to express halorhodopsin (AAV5-EF1a-DIO-eNpHR3.0-eYFP; DIO-eNpHR) on dSPNs (*Figure 4A,D, and E*) or iSPNs (*Figure 4A, F, and G*) so as to allow the selective inhibition of each pathway using light from a 625 nm wavelength LED delivered to the pDMS. Using ex vivo slice electrophysiology, we validated the effectiveness of this inhibition. Following the delivery of LED light, action potential frequency of SPNs (dSPNs and iSPNs were pooled due to little or no difference in their response to LED illumination) was significantly reduced in DIO-eNpHR labelled SPNs (*Figure 4B and C*; pairwise comparison, F(1,8)=11.76, p=0.009) but not in non-labelled SPNs (*Figure 4—figure supplement 1A and B*; pairwise comparison, F(1,4)=1.4, p=0.306). Confocal images of a representative recorded dSPN, iSPN, and an unlabelled SPN are presented in *Figure 4—figure supplement 1C–E*.

Groups used in this experiment are summarized in *Figure 4H*. Rats in the eYFP control group expressed a control fluorophore (AAV5-EF1a-DIO-eYFP; DIO-eYFP) in either dSPNs or iSPNs, serving as a control for LED light delivery. Due to the lack of eNpHR construct, this group of rats retained intact pathway function and, indeed, as there were no differences in their press rates during training

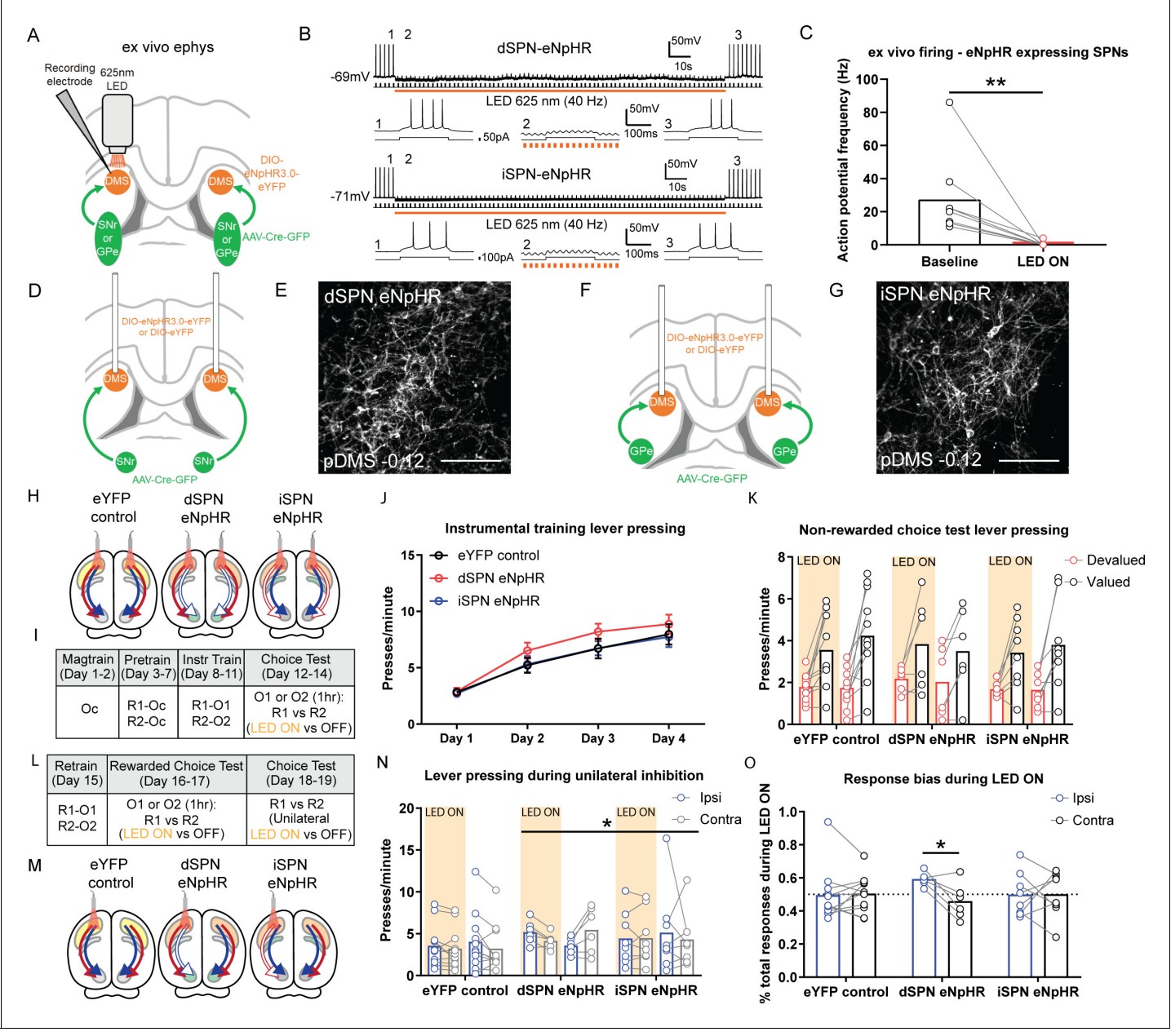

**Figure 4.** Bilateral optogenetic inhibition of direct spiny projection neurons (dSPNs) or indirect spiny projection neurons (iSPNs) leaves the expression of goal-directed learning intact but unilateral inhibition of dSPNs biases performance. (**A**) Schematic depicting the design for ex vivo recordings; retro-Cre was injected into the substantia nigra pars reticulata (SNr) or globus pallidus (GPe) and DIO-eNpHR into the posterior dorsomedial striatum (pDMS); direction of arrows indicates retrograde transport of the virus. (**B**) Example trace from one dSPN (top) and iSPN (bottom) expressing eNpHR recorded under current-clamp with an injection of +50 or +100 pA current (200 ms, 0.5 Hz). Orange bar denotes time period when 625 nm LED was applied to the neuron (40 Hz, 150 s). Bottom three panels of each SPN are expanded traces from the above at time points before (1), during (2), and after (3) LED illumination. (**C**) Action potential frequency of eNpHR-labelled SPNs (four dSPNs and five iSPNs) at baseline (10 s period before LED) versus LED (10 s period from beginning of LED). (**D**) Surgery design for targeting dSPNs; retro-Cre was injected bilaterally into the SNr and DIO-eNpHR or DIO-eYFP was injected bilaterally into the pDMS. Cannulae were inserted bilaterally into the pDMS. (**E**) Confocal image (scale bar, 100 μm) showing DIO-eNpHR expression in pDMS dSPNs. (**F and G**) Same as D and E but for pDMS iSPNs; retro-Cre was injected into the GPe. (**H**) Summary of experimental groups; blue arrows indicate intact direct pathway, red arrows indicate intact indirect pathway, and unfilled arrows indicate inhibited pathway. (**I**) Summary of the experimental design; R1 and R2 indicate left and right lever responses; Oc, O1, and O2 indicate distinct food outcomes; LED ON versus OFF indicates training or test days when LED light was delivered. (**J**) Mean (± SEM) lever presses per minute averaged across each day of instrumental training for each group. (**K**) Mean presses per minute on the devalued and valued lever for each rat in each group, averaged across two tests, during LED ON (orange shaded) and LED OFF (non-shaded) periods. (**L**) Continuation of I. (**M**) Same as H but showing unilateral LED manipulation. (**N**) Mean presses per minute on the lever ipsilateral and contralateral to unilateral inhibition for each rat in each group during periods of

*Figure 4 continued on next page*

Figure 4 continued

LED ON (orange shaded) and LED OFF (non-shaded), averaged across two tests with inhibition in each hemisphere. (O) Mean proportion of responding on the ipsilateral and contralateral lever during the LED ON period, as a proportion of the total responding on each respective lever, for each animal in each group, averaged across two tests. For all data, bars represent group means. *p<0.05, **p<0.01.

The online version of this article includes the following figure supplement(s) for figure 4:

**Figure supplement 1.** Verification of cellular expression and non-labelled SPN activity for the optogenetic inhibition experiment.

**Figure supplement 2.** Verification of the eNpHR virus and supplemental behavioral data related to the optogenetic inhibition experiment.

and no differential effects of LED light presentation on test (Fs <1), they were pooled into a single control group. Example images of retro-Cre expression and the extent of virus spread in the SNr and GPe are presented in *Figure 4—figure supplement 2A–D*. We quantified the mean number of cells in the pDMS expressing DIO-eYFP or DIO-eNpHR (*Figure 4—figure supplement 2E*); animals that received DIO-eYFP had greater expression than those that received DIO-eNpHR (F(1,22)=12.8, p=0.002), but there were no differences between Groups dSPN eNpHR and iSPN eNpHR, nor any differences between hemispheres in any group (Fs <1.5, lowest p=0.234). The spread of pDMS virus expression and location of optogenetic cannula tips for each group are presented in *Figure 4—figure supplement 2F*.

A summary of the behavioural design is shown in *Figure 4I*. Instrumental pre-training and training sessions were conducted without LED light delivery. For analysis of the optogenetic inhibition experiments, we compared the control group against the two experimental groups, dSPN eNpHR and iSPN eNpHR as well as comparing Groups dSPN eNpHR and iSPN eNpHR against each other. All groups acquired the instrumental response across pre-training (*Figure 4—figure supplement 2G*). Across four days of instrumental training (*Figure 4J*), all groups increased their press rates (F(1,22)=188.35, p<0.001) and there were no differences between groups in overall response rates (all Fs <1.1). We tested that actions were goal directed using outcome devaluation followed by a choice extinction test. The test was 7.5 min long; dSPNs or iSPNs in the pDMS were inhibited with orange LED light during the first and last 2.5 min, separated by 2.5 min with no LED light. Responding in LED ON periods was similar (no group × LED ON period, lever × LED ON period or group × lever × LED ON period interactions; all Fs <1, lowest p=0.623) and so averaged for comparison with the LED OFF period. We found a main effect of devaluation (*Figure 4K*; F(1,22)=37.57, p<0.001) but no effect of LED and no group × LED × lever interaction (Fs <1, lowest p=0.390), indicating that rats showed intact outcome devaluation, and this was unaffected by optogenetic inhibition of either dSPNs or iSPNs. Additionally, optogenetic inhibition, whether of dSPNs or iSPNs, had no effect on the rats' ability to discriminate between outcomes or levers during a rewarded devaluation choice test (*Figure 4—figure supplement 2H*; main effect of lever [valued versus devalued] F(1,22)=29.28, p<0.001; no group × lever, group × LED or group × LED × lever interactions, highest F(1,22)=2.87, p=0.104 [LED × lever interaction]). Therefore, unlike goal-directed learning, bilateral optogenetic inhibition of dSPNs had no effect on goal-directed performance.

## Unilateral inhibition of dSPNs in the pDMS induces an ipsilateral response bias

To further examine the role of dSPNs and iSPNs in performance we used the same animals and examined the effect of unilateral inhibition of these SPNs on response bias in a choice extinction test conducted without devaluation (i.e. without pre-feeding; *Figure 4L*). As represented in *Figure 4M,* a single patch cord was attached to either the left or right cannula (first and second tests, respectively) and rats were again tested for 7.5 min as described previously. There were no differences between the first and last LED ON periods and so these blocks were pooled and compared to the LED OFF block. Test data are presented in *Figure 4N*. There were no differences in press rates according to lever side or LED, averaged across group, nor between Group eYFP and the two eNpHR groups (averaged) in these measurements (F(1,22)=1.67, p=0.210). There was however a significant group × lever × LED interaction between Group dSPN eNpHR and Group iSPN eNpHR (F(1,22)=5.78, p=0.025), indicating that the effect of the light on lever choice differed between these groups. To follow up, we analysed each of the groups separately, comparing responding on the ipsilateral or contralateral levers during the LED ON period as a ratio of total responding (LED ON plus

LED OFF periods) for each lever (*Figure 4O*). For rats in Group iSPN eNpHR and Group eYFP control, responses on each lever during the LED ON period were ~50% of the total responding on each lever, whereas for rats in Group dSPN eNpHR we observed a significant ipsilateral bias (F(1,22) =6.28, p=0.02); 59% versus 46% for ipsilateral versus contralateral levers, respectively. Therefore, unilateral dSPN inhibition, but not iSPN inhibition, produced an ipsilateral bias in goal-directed performance. To ensure that this effect was not attributable to LED-induced rotational behaviour, we quantified rotations for each animal during this test (*Figure 4—figure supplement 2I and J*) and found no effect of unilateral LED light delivery on the number of ipsilateral (Fs <2.1, lowest p=0.169) or contralateral (Fs <1, lowest p=0.342) rotations in any group.

Given this finding, we revisited the Zif268 quantification described in our initial experiment, separating expression in each hemisphere based on the location of the lever in the chamber (*Figure 4—figure supplement 2K and L*); i.e., if a rat had the lever on the right side of the chamber, the left hemisphere was designated as 'contralateral', and the right hemisphere as 'ipsilateral'. For Zif268 expression in dSPNs, there was a pronounced hemispheric difference in Group Instrumental but not in Group Yoked; rats in Group Instrumental had significantly more Zif268 in the hemisphere contralateral to the lever side (group × hemisphere interaction, F(1,20)=4.84, p=0.040). In contrast, analysis of Zif268 expression in iSPNs revealed no significant differences between hemispheres (no group × hemisphere interaction, F(1,20)=0.70, p=0.413). Therefore, acquisition of goal-directed actions appeared to produce a lateralized (contralateral to action space) increase in the activity of dSPNs sufficient to bias instrumental performance.

## Optogenetic inhibition of iSPNs, but not dSPNs, in the pDMS impairs flexible action selection in response to changes in the action-outcome contingency

Our experiments so far have failed to detect any significant role for iSPNs in the acquisition and performance of goal-directed actions. In fact, recent findings suggest that iSPN activity may only be necessary for learning and performance when instrumental contingencies change (*Matamales et al., 2020*). In order to assess this possibility, we examined the effect of dSPN and iSPN inhibition when action-outcome contingencies were altered using a series of reversals in a two-choice situation. We used an instrumental reward reversal paradigm, wherein the rats were exposed to the same action-outcome contingencies as initial training, but with one lever rewarded and the other lever non-rewarded (i.e., if R1→O1 then R2→Ø; whereas if R2→O2 then R1→Ø). These contingencies were reversed every 2.5 min in semi-random fashion (with no more than two of the same trial type presented sequentially). Each test contained 10 × 2.5 min trials, and rats were tested each day for 3 days. LED light was delivered bilaterally during all trials, but not between trials (*Figure 5A and B*).

Press rates on the non-rewarded lever, the rewarded lever, and the difference in press rates between the two levers are presented in *Figure 5C–E*. There were significant effects of training day in all three metrics, indicating that rats decreased responding on the non-rewarded lever across days (*Figure 5C*; F(1,22)=9.84, p=0.005) and increased responding on the rewarded lever (*Figure 5D*; F(1,22)=11.61, p=0.003), and that the magnitude of the difference between the rewarded and non-rewarded levers increased across days (*Figure 5E*; F(1,22)=19.82, p<0.001). We compared the difference in responding on the rewarded and non-rewarded levers between groups on each day (*Figure 5E*). On Day 1 there was a significant difference between Group eYFP control and the two eNpHR groups (F(1,22)=7.03, p=0.015), which was driven by reduced discrimination in Group iSPN eNpHR relative to the other groups. This was confirmed by a significant difference between Group iSPN eNpHR and Group dSPN eNpHR (F(1,22)=4.39, p=0.048). The difference between the control group and the two eNpHR groups was maintained on Day 2 (F(1,22)=5.91, p=0.024), but not between the two eNpHR groups (F(1,22)=1.8, p=0.193). There were no significant differences on Day 3 (Fs <1, lowest p=0.548). To confirm that the observed effects were primarily driven by rats in Group iSPN eNpHR, we conducted a follow-up analysis comparing each eNpHR group to the control group. On Days 1 and 2, rats in Group iSPN eNpHR showed a significant deficit in discriminating the rewarded versus the non-rewarded levers as determined by the difference in the press rates on these levers (Bonferroni k=6, Day 1, F(1,22)=12.45, p=0.002; Day 2, F(1,22)=8.4, p=0.008). This deficit reduced over days and was not detectable on Day 3 (Fs <1). There was no difference in performance between Group dSPN eNpHR and the eYFP control on any day (Fs <1.6).

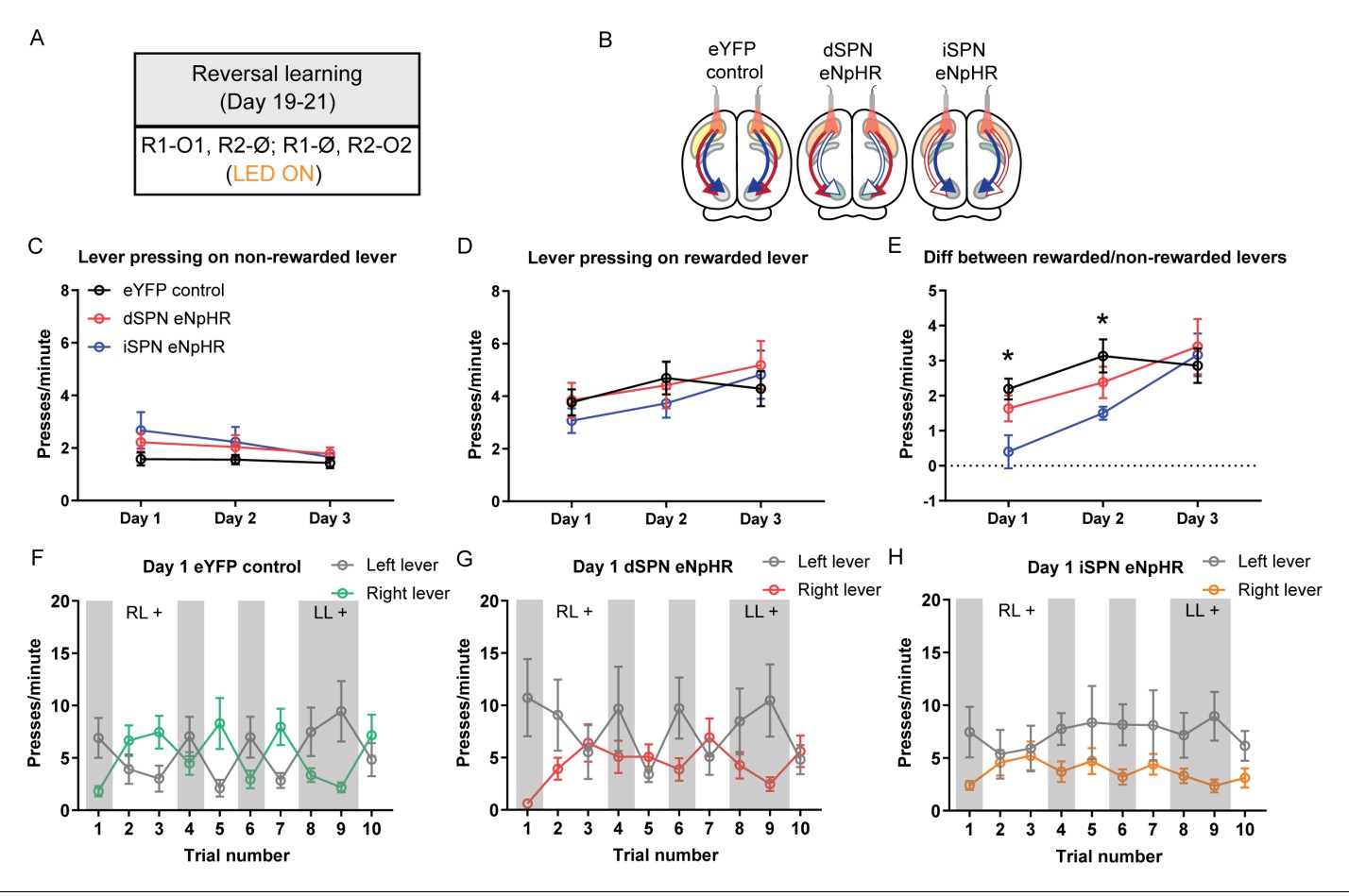

**Figure 5.** Optogenetic inhibition of indirect spiny projection neurons, but not direct spiny projection neurons, in the posterior dorsomedial striatum impairs response flexibility. (A) Summary of the experimental design; R1 and R2 indicate left and right lever responses; O1 and O2 indicate distinct food outcomes; Ø indicates non-rewarded responses; LED light was delivered during all lever presentations. (B) Summary of experimental groups; blue arrows indicate intact direct pathway function, red arrows indicate intact indirect pathway function, and unfilled arrows indicate inhibited pathway. (C) Mean (± SEM) presses per minute on the non-rewarded lever across 3 days of reversal training in all groups. (D) Mean (± SEM) presses per minute on the rewarded lever across 3 days of reversal training in all groups. (E) Mean (± SEM) difference in press rate (expressed as presses per minute) between the rewarded and non-rewarded levers for each group. (F–H) Mean (± SEM) lever presses per minute on the left lever and right lever for rats in each group on Day 1 of training, averaged across each 2.5 min trial – grey-shaded regions indicate trials in which the left lever was rewarded (LL+) and non-shaded regions indicate trials in which the right lever was rewarded (RL+). *p<0.05.

The online version of this article includes the following figure supplement(s) for figure 5:

**Figure supplement 1.** Days 2 and 3 of reversal training in the optogenetic inhibition experiment.

Figure 5F–H and Figure 5—figure supplement 1A–F show the rate of responding on the left and right levers for each group, across the 3 days of training – shaded blocks indicate those in which the left lever was rewarded and unshaded those in which the right lever was rewarded. We analysed each day separately: Rats in Group eYFP control and dSPN eNpHR adjusted their performance according to the positive contingency during each epoch on Day 1 (Figure 5F–H); there was a significant main effect of reversal (lever × trial, Bonferroni, k=9, $F_{(1,22)}$=56.4, p<0.001) and this did not differ between groups ($F_{(1,22)}$=1.28, p=0.27). However, rats in Group iSPN eNpHR failed to adjust their performance as the positive contingency shifted across epochs; there was a significant Group × reversal interaction between Groups iSPN eNpHR and eYFP control (Bonferroni, k=9, $F_{(1,22)}$=12.27, p=0.002) indicating that Group iSPN eNpHR was impaired at flexibly adjusting performance on Day 1. This pattern was partially maintained on Day 2 (Figure 5—figure supplement 1A–C): There was a significant main effect of reversal (Bonferroni, k=9, $F_{(1,22)}$=87.36, p<0.001); this did not differ between Group dSPN eNpHR and eYFP control ($F_{(1,22)}$=1.51, p=0.232) and there was a substantial

but non-significant Group × reversal interaction between Group iSPN eNpHR and eYFP control (Bonferroni, k=9, F(1,22)=8.39, p=0.008). Finally, on Day 3 (*Figure 5—figure supplement 1D–F*), groups maintained their sensitivity to contingency (main effect of reversal, Bonferroni, k=9, F(1,22) =77.83, p<0.001), and there were no differences in the magnitude of reversal between Group dSPN eNpHR and eYFP control (F(1,22)=0.37, p=0.548), nor between Group iSPN eNpHR and eYFP control (F(1,22)=0.16, p=0.697). Therefore, optogenetic inhibition of iSPNs impaired rats' ability to adjust their instrumental performance flexibly in accordance with changes in the action-outcome contingency during early reversal learning, consistent with the claim that iSPNs are necessary to flexibly update instrumental learning (*Matamales et al., 2020*).

## Discussion

Using a circuit-specific approach to examine the involvement of the direct and indirect pathway projection neurons in goal-directed learning, performance, and action updating, we found evidence that goal-directed learning recruits dSPNs, but not iSPNs, that the role of this recruitment for performance is specific to the orientation of the response in contralateral space, and, as a consequence, that unilateral inhibition of dSPNs, but not iSPNs, produced an ipsilateral response bias. Furthermore, despite a lack of involvement in initial learning and performance, we found that iSPNs, but not dSPNs, were necessary for flexibly updating goal-directed performance when the currently rewarded action-outcome contingency changes.

### SPNs and goal-directed learning: A specific function of the direct pathway

Prior research assessing the involvement of dSPNs and iSPNs in goal-directed learning has generally inferred a greater relative involvement of dSPNs based on the use of cell-type specific GFP expression in either D1-GFP (*Maroteaux et al., 2014*; *Shan et al., 2014*) or A2A/D2-GFP (*Shan et al., 2014*; *Matamales et al., 2020*) mice. We confirmed this involvement using the circuit-specific approach described here, both in the recruitment of Zif268 signalling, which was specifically increased in dSPNs and not iSPNs following instrumental training, and using chemogenetic inhibition, where we found causal evidence that dSPNs but not iSPNs are required for initial goal-directed learning. Furthermore, this effect of dSPN inhibition on goal-directed learning was specific to dSPNs and was not due to changes in the relative balance between the two populations of pDMS projection neurons; chemogenetic stimulation of iSPNs during the same period of training had no effect on goal-directed learning.

In a functional sense, previous studies conducting online assessments of instrumental conditioning have told us several things regarding the role of SPNs in instrumental performance in relation to reward history (e.g., *Tai et al., 2012*). However, what such assessments cannot determine is whether the actions being observed are in fact goal directed. In order to make that assertion, tests are required to demonstrate that subjects have encoded the relationship between actions and their consequences and are sensitive to the current value of the outcome (*Balleine, 2019*). Tests for outcome devaluation under extinction conditions provide prima facie evidence for both criteria; when animals bias responding away from a devalued lever, they demonstrate awareness of the action-outcome relationships, and the capacity to modulate their choice based on the current value of each outcome in the absence of any feedback to inform this choice (*Balleine and Dickinson, 1998*). Importantly, the current experiments established the necessary functional role of dSPNs (rather than iSPNs) in goal-directed learning for actions verified to be goal directed.

### SPNs and goal-directed performance: Involvement of the direct pathway

We have previously argued (*Peak et al., 2019*) that glutamatergic inputs to the dorsal striatum constitute 'learning pathways' in the broader basal ganglia circuitry, and striatal output pathways constitute 'performance pathways'. Central to that argument is that the dorsal striatum constitutes an interface between these two functions and the current results refine this, suggesting that both goal-directed learning and performance rely specifically on dSPN input and output processes in the pDMS.

As has been reported previously, we found an instrumental training-induced increase in activity-related signalling in pDMS dSPNs. However, here we assessed the activity of each hemisphere separately relative to the position of the lever and found that this elevation was response specific; that is, it was lateralized with respect to the position of the manipulanda for the trained response. This is likely related to performance factors. Thus, for example, the activity of dopamine neurons projecting from the SNc to the DMS has been reported to be lateralized according to response type; when rats press a left lever, SNc-to-DMS dopamine neurons in the right hemisphere have been found to increase their activity independently of action value (*Parker et al., 2016*; *Lee et al., 2019*). These results predict that the lateralization of activity in dSPNs during goal-directed performance is driven by a lateralized dopamine input converging with the bilateral input from prefrontal cortex (*Hart et al., 2018a*; *Hart et al., 2018b*).

Although previous studies have reported an effect of iSPN inhibition on goal-directed performance (*Carvalho Poyraz et al., 2016*), there have been no studies assessing the role of dSPNs, and indeed, it was our prediction that this performance would be reliant on dSPN activity. Instead, we found that bilateral optogenetic inhibition, whether of dSPNs or iSPNs, had no effect on choice performance on test. Nevertheless, in the same animals, we found that unilateral optogenetic inhibition of dSPNs was effective in inducing an ipsilateral response bias (relative to the hemisphere of inhibition) in goal-directed responding, whereas unilateral iSPN inhibition had no effect. This effect of unilateral, but not bilateral inhibition of dSPNs on goal-directed performance emphasizes the importance of lateralized dSPN activity in inducing this response bias. By bilaterally dampening dSPN output, we did nothing to the relative balance of dSPN activity in each hemisphere and, as such, any existing response bias was unchanged. Lateralized control of motor output by dSPNs has been well demonstrated; unilateral activation or suppression biases contralateral or ipsilateral movements, respectively (*Kravitz et al., 2010*; *Bay Kønig et al., 2019*), whereas unilateral stimulation of dSPNs produces a contralateral bias in lever pressing (*Tai et al., 2012*). Our results indicate that these findings extend to goal-directed actions, consistent with the finding that DA terminals in the DMS respond more strongly during contralateral choices than ipsilateral choices and that this activity, too, precedes action performance (*Parker et al., 2016*; *Lee et al., 2019*). What appears to be critical here is the location of the lever relative to the magazine; lever pressing and magazine checking necessitate both ipsilateral and contralateral movements (relative to the hemisphere of recording, stimulation, or inhibition) and this is likely the source of the increased bias (*Lee et al., 2019*).

Nevertheless, the failure to find an effect of bilateral dSPN inhibition on performance was surprising. Previous studies have found that lesions or pharmacological manipulations of pDMS conducted after training are as effective as lesions made prior to training in abolishing the influence of outcome devaluation on performance (e.g., *Yin et al., 2005a*; *Yin et al., 2005b*; *Shiflett et al., 2010*). The current approach induced a far more specific and subtle reduction in neuronal output to either the SNr or GPe, and it is possible that unlike goal-directed learning, performance can be maintained with dramatically reduced (but not abolished) pathway function. Alternatively, such performance may rely instead on some other, as yet unspecified pathway. Indeed, such findings open up for consideration alternative ways in which the pDMS alters motor performance and the means by which striatal feedback circuits are integrated to that end, a task that awaits future research.

## SPNs and action flexibility: A role for the indirect pathway

Despite the lack of involvement of iSPNs in learning, we found clear evidence that bilateral inhibition of iSPNs, but not dSPNs, in the pDMS impaired the flexible updating of the contingency controlling goal-directed performance in rats exposed to a series of reversals. A previous study using nose-poke reversals in mice reported effects of inhibition of both dSPNs and iSPNs on reversal, arguing, using a modelling analysis, that these effects reflected differential involvement of dSPNs in learning action values and of iSPNs in updating those values (*Kwak and Jung, 2019*). That study, however, used D1- and D2-Cre mice with hM4D-DREADDs to inactivate dSPNs and iSPNs infused into a central region of the dorsal striatum encompassing both medial and lateral regions. The dorsal striatum is highly segregated both in terms of its cortical inputs (*Hunnicutt et al., 2016*) and the functional role of its sub-regions in instrumental actions, with the most robust amongst these distinctions being the specific role of the pDMS, and not the more lateral (or indeed anterior) regions of the dorsal striatum, in goal-directed actions (*Balleine, 2019*). It is also unclear whether the reported effects were specific to iSPNs or were mediated by other D2-receptor expressing neurons, particularly various

interneurons that also express D2 receptors. It is difficult, therefore, to apply the findings of this previous study to formulate predictions for our assessment. Here we found evidence that when iSPNs in the pDMS were inhibited, rats were slower to learn which of two levers was rewarded and to switch their responding appropriately. Importantly, this failure was overcome with continued training, suggesting that, with sufficient additional experience, rats were capable of learning response strategies to compensate for this impairment.

A lack of flexibility in this task could be due to several factors but, primarily, impairments can manifest as an inability to suppress responding on the non-rewarded lever (perseveration), or a failure to increase responding on the rewarded lever. In fact, we found that the impairment induced by iSPN inhibition was related to both of these factors and, indeed, there is recent evidence to suggest that both of these forms of response flexibility require iSPNs. For example, *Matamales et al., 2020* demonstrated that ablating iSPNs in the DMS encouraged recurrent responding during the extinction of goal-directed learning (response perseveration), whereas *Nonomura et al., 2018* reported that iSPNs in the DMS are sensitive to non-reward and, when optogenetically stimulated following a non-rewarded-response, promote switching to an alternate response.

Although, given prediction error analyses, it might be supposed that outcome omission forms the basis for iSPN involvement in these forms of reversal and extinction learning, *Matamales et al., 2020* found similar involvement of iSPNs when animals were exposed to reversal of the identity of the outcomes of two otherwise continuously rewarded actions. It seems reasonable to conclude, therefore, that the causal evidence for the involvement of iSPNs in contingency reversal observed in the current study supports a role for iSPNs in updating the currently (non) rewarded action-outcome contingency. From this perspective iSPNs guide plasticity associated with changes in reward and non-reward so as to interleave new learning with prior encoding in a manner that minimizes interference between them.

Importantly, other evidence suggests that action-outcome updating is not solely dependent on dopamine-related signalling at iSPNs and relies on other sources of local modulation, most notably acetylcholine that is delivered locally via giant aspiny cholinergic interneurons. Cholinergic activity in the striatum is tightly bound to dopamine activity (*Threlfell et al., 2012*) and is critical for shaping goal-directed learning in the striatum (*Bradfield et al., 2013*; *Matamales et al., 2016*). Furthermore, recent evidence suggests that iSPNs and CINs interact to orchestrate striatal circuit function (*Tanimura et al., 2019*) under the control of inputs from the parafascicular thalamus via the thalamostriatal pathway (*Bradfield et al., 2013*; *Tanimura et al., 2019*). It is possible that both the Pf-pDMS pathway and iSPNs contribute to this in a serial manner; thalamic inputs generate burst-pause firing in pDMS CINs which evokes a prolonged postsynaptic excitability in iSPNs but not dSPNs, driven by the activation of M1 muscarinic receptors (*Ding et al., 2010*). Thus, thalamic activation of pDMS CINs creates a temporal window during which the striatal network is biased towards the activation of neighbouring iSPNs. Given this account, it should be anticipated that a loss in the function of either aspect of this local circuit will be sufficient to abolish the capacity of animals to update goal-directed learning.

## Conclusions

The control of instrumental actions according to our current goals requires the differential recruitment of dSPN and iSPN pathways. By assessing each pathway according to projection target, rather than receptor expression, we were able to dissociate their functions in goal-directed actions along several lines. Broadly, dSPNs were found to be important for encoding goal-directed learning whereas goal-directed performance is reflected in, and modulated by, the asymmetrical activity of dSPNs in each hemisphere and the resulting drive of responses in contralateral action space. By contrast, when contingencies change, iSPNs are required to update those contingencies to support response flexibility.

## Materials and methods

**Key resources table**

*Continued on next page*

*Continued*

| Reagent type (species) or resource | Designation | Source or reference | Identifiers | Additional information |
|---|---|---|---|---|
| Reagent type (species) or resource | Designation | Source or reference | Identifiers | Additional information |
| Strain, strain background (*Rattus norvegicus*, Long-Evans) | Long-Evans Rat | Randwick Rat Breeding Facility, Randwick, NSW, Australia | NA | Wild-type rats |
| Strain, strain background (*Rattus norvegicus*, Long-Evans) | Long-Evans Rat | Animal Resources Centre, Perth, WA, Australia | NA | Wild-type rats |
| Antibody | Anti-goat Alexa Fluor 647 (donkey polyclonal) | Invitrogen | Cat# A-21447; RRID:AB_2535864 | (1:1000) |
| Antibody | Anti-mouse Alexa Fluor 488 (donkey polyclonal) | Invitrogen | Cat# A-21202; RRID:AB_141607 | (1:1000) |
| Antibody | Anti-mouse Alexa Fluor 546 (donkey polyclonal) | Invitrogen | Cat# A10036; RRID:AB_2534012 | (1:1000) |
| Antibody | Anti-rabbit Alexa Fluor 488 (donkey polyclonal) | Invitrogen | Cat# A-21206; RRID:AB_2535792 | (1:1000) |
| Antibody | Anti-rabbit Alexa Fluor 546 (donkey polyclonal) | Invitrogen | Cat# A10040; RRID:AB_2534016 | (1:1000) |
| Antibody | Anti-cholera-toxin B subunit (goat polyclonal) | LIST Biological Laboratories Inc | Cat# 703; RRID:AB_10013220 | (1:2000) |
| Antibody | Anti-DARPP-32 (mouse monoclonal) | BD Biosciences | Cat# 611520; RRID:AB_398980 | (1:1000) |
| Antibody | Anti-EGR1 (rabbit polyclonal) | Santa Cruz Biotechnology | Cat# C-19; RRID:AB_2231020 | (1:300) |
| Antibody | Anti-GFP (rabbit polyclonal) | Invitrogen | Cat# A-11122; RRID:AB_221569 | (1:1000) |
| Antibody | Streptavidin, Alexa Fluor 488 | Invitrogen | Cat# S11223; RRID:AB_2315383 | (1:500) |
| Recombinant DNA reagent | AAV-Cre: rAAV5.CMV.HI. eGFP-CRE.WPRE.SV40 | AddGene | Cat# 105545 | |
| Recombinant DNA reagent | ChR2: AAV5-CaMKIIa-hChR2(H134R)-eYFP | AddGene | Cat# 26969 | |
| Recombinant DNA reagent | Cre-eNpHR: AAV5-Ef1a-DIO-eNpHR3.0-eYFP | AddGene | Cat# 26966 | |
| Recombinant DNA reagent | Cre-eYFP: AAV5-Ef1a-DIO-eYFP | AddGene | Cat# 27056 | |
| Recombinant DNA reagent | Cre-hM3D: rAAV5/hSyn-DIO-hM3D-mCherry | AddGene | Cat# 44361 | |
| Recombinant DNA reagent | Cre-hM4D: rAAV5/hSyn-DIO-hM4D(Gi)-mCherry | AddGene | Cat# 44362 | |
| Recombinant DNA reagent | Cre-hM4D: rAAV5/hSyn-DIO-hM4D(Gi)-mCherry | UNC Vector Core | SCR_002448 | |
| Recombinant DNA reagent | Cre-mCherry: rAAV5/hSyn-DIO-mCherry | AddGene | Cat# 50459 | |
| Chemical compound, drug | Clozapine-N-oxide | RTI International | NA, Batch ID: 13626–76 | |
| Software, algorithm | GraphPad Prism | GraphPad Software Inc | RRID:SCR_002798 | |
| Software, algorithm | Image J | NIH | RRID:SCR_003070 | |
| Software, algorithm | Med-PC program | Med Associates | RRID:SCR_012156 | |
| Software, algorithm | PSY statistical program | School of Psychology, UNSW | NA | |

*Continued on next page*

*Continued*

| Reagent type (species) or resource | Designation | Source or reference | Identifiers | Additional information |
|---|---|---|---|---|
| Other | AMCA Avidin D | Vector Laboratories | Cat# A-2008 | |
| Other | Biocytin | Sigma-Aldrich | Cat# B1758 | |
| Other | Cholera-toxin B subunit | List Biological Laboratories | Cat# 104; RRID:AB_2313636 | |
| Other | Fluorogold | Fluorochrome | NA | |

## Subjects

Subjects for all experiments were Long-Evans rats, obtained either from the Animal Resource Centre (Perth, Western Australia) or from the Randwick Rat Breeding Facility (Randwick, New South Wales). All rats were healthy and experimentally naïve at the beginning of each experiment and at least 8 weeks old prior to surgery. Rats were housed in transparent, plastic boxes with two to four rats per box in a climate-controlled colony room and maintained on a 12 hr light/dark cycle (lights on at 07:00 hr). All experiments were conducted during the light phase. Water and standard lab chow were available ad libitum prior to the start of experiments. For all experiments, rats were randomly assigned to experimental groups and sample sizes were based on experiments previously conducted and reported (e.g., *Hart et al., 2018b*). Each experiment was conducted once and so the same cohort of animals was not tested multiple times (technical replication), nor was the same experiment repeated on multiple occasions (biological replication). All experiments conformed to the guidelines on the ethical use of animals maintained by the *Australian code for the care and use of animals for scientific purposes,* and all procedures were approved by the Animal Care and Ethics Committee at either the University of New South Wales or the University of Sydney.

### SPN tracing and immunofluorescence experiment
Subjects were 25 experimentally naïve female outbred Long-Evans rats obtained from the Animal Resource Centre (Perth, Western Australia).

### Electrophysiology recordings
Subjects for recordings of SPNs were 24 (four for Group CNO control, five for Group dSPN hM4D, five for Group iSPN hM4D, three for Group iSPN hM3D, four for Group dSPN halorhodopsin, and three for Group iSPN halorhodopsin) experimentally naïve female Long-Evans rats obtained from the Randwick Rat Breeding Facility (Randwick, New South Wales).

### DREADDs suppression experiment
Subjects were 79 experimentally naïve Long-Evans rats (36 males and 43 females) obtained from either the Animal Resource Centre (Perth, Western Australia) or from the Randwick Rat Breeding Facility (Randwick, New South Wales).

### DREADDs stimulation experiment
Subjects were 30 experimentally naïve Long-Evans rats (14 males and 16 females) obtained from the Randwick Rat Breeding Facility (Randwick, New South Wales).

### Optogenetic inhibition experiment
Subjects were 39 experimentally naïve Long-Evans rats (13 males and 24 females) obtained from the Randwick Rat Breeding Facility (Randwick, New South Wales).

## Stereotaxic surgery
### Surgical procedures
Rats received stereotaxic surgery conducted under isoflurane gas anaesthesia (0.6 l/min; isoflurane at 5% during induction and 2–2.5% maintenance). Animals were placed in a stereotaxic frame (Kopf instruments) and a subcutaneous injection of bupivacaine hydrochloride at the incision site and a

subcutaneous injection of carprofen in the lower flank administered. An incision was made to expose the scalp, membrane on top of skull was cleared, and the head was adjusted to align bregma and lambda on the same horizontal plane. Small holes were drilled in the skull above the target structures and a Hamilton syringe (1.0 µl) connected to an Infuse/Withdraw pump (Harvard Apparatus) or a capillary micropipette connected to a Nanoject (Drummond Scientific) was lowered into the brain for infusions of viruses or tracers, respectively. Following each infusion, the needle was left in position for 3–5 min to allow diffusion. Following the final infusion, the incision site was closed and secured with surgical staples. Rats then received an injection of antibiotic (either Benacillin [0.3 ml i. p.] or Duplocillin [0.15 ml/kg s.c.]) and saline (3 ml i.p.). Rats were given a minimum of 2 weeks recovery time following surgery to allow for sufficient tracer expression, or 4 weeks for viral expression.

Surgical co-ordinates for each region were pre-determined from pilot studies and varied slightly between experiments. All infusions were bilateral. Retrograde tracers or retro-Cre were infused into the GPe or SNr at the following coordinates (mm from bregma): males: GPe: AP −1.25; ML ± 3.1; DV −6.9; SNr: AP −5.8; ML ± 1.95; DV −8.8, females: GPe: AP −1.1; ML ± 3.0; DV −6.9; SNr: AP −5.6 to −5.8; ML ± 1.9; DV −8.6 to −8.65. Channelrhodopsin was infused into the PL at the coordinates: AP +2.9; ML ± 0.6; DV −3.8. DREADDs, halorhodopsin, or control fluorophores were infused into the pDMS at the following coordinates: males: AP −0.3; ML ± 2.35 to ±2.40; DV −4.7, females, AP −0.25; ML ± 2.3 to ±2.35; DV −4.5 to −4.6. For rats in optogenetic inhibition experiment, glass cannulae were inserted above the pDMS at the following coordinates: males: AP −0.3; ML ± 2.40 to ±2.45; DV −4.50, females: AP −0.25; ML ± 2.35 to ±2.40; DV −4.35 to −4.40.

## Viruses and tracers

SPNs were targeted by infusing the retrograde tracers CTB (1% in dH$_2$O; List Biological Laboratories) and FG (3% in saline; Fluorochrome) into the GPe and SNr, respectively. Both tracers were infused at a total volume of 82.2 nl and a rate of 23 nl/min. Volumes were selected based on pilot studies designed to contain the spread of tracers within each region.

Projection neurons in the PL were targeted by infusing Channelrhodopsin, AAV5-CaMKIIa-hChR2 (H134R)-EYFP (ChR2-eYFP) into the PL at a total volume of 300 nl and a rate of 150 nl/min.

Striatal SPNs were targeted by infusing retro-Cre (rAAV5.CMV.HI.eGFP-CRE.WPRE.SV40) into either the SNr (for dSPNs) or GPe (for iSPNs). Retro-Cre was infused into the SNr at a total volume of between 300 nl and 500 nl at a rate of 100 nl/min, and into the GPe at a total volume of 150 nl and a rate of 50 nl/min.

All DREADDs viruses, DIO-hM4D (rAAV5/hSyn-DIO-hM4D(Gi)-mCherry), DIO-mCherry (rAAV5/hSyn-DIO-mCherry), and DIO-hM3D (rAAV5/hSyn-DIO-hM3D-mCherry), were infused into the pDMS at a total volume of 750 nl at a rate of 100–150 nl/min.

For optogenetic inhibition, 500 nl of DIO-eNpHR (AAV5-EF1a-DIO-eNpHR3.0-eYFP) or DIO-eYFP (AAV5-EF1a-DIO-eYFP) was infused into the pDMS at a rate of 100 nl/min.

For all virus and tracer infusions, the needle was left in place for a minimum of 3 min following each infusion.

## Behavioural procedures

### Apparatus

For immunofluorescence and DREADDs experiments, training and testing were conducted in 16 Med Associates operant chambers, individually housed in light and sound attenuating cabinets. Each chamber was fitted with a food magazine connected to a 45 mg grain pellet (Bioserve Biotechnologies) dispenser as well as two pumps fitted with external syringes that delivered either 0.2 ml of 20% sucrose solution (white sugar, Coles, Australia) or 0.2 ml of 20% maltodextrin solution (Poly-Joule, Nutrica, Australia). An infrared detector was situated horizontally across the inside of the magazine to detect head entries.

The chambers were also fitted with two retractable levers, located on either side of the magazine. A house light (3W, 24V) was located in a central position at the top of the wall opposite to the food magazine and illuminated during all experimental stages, unless otherwise stated. Training and testing sessions were pre-programmed and controlled by computers external to testing rooms, using Med Associates software (Med-PC IV or V), which also recorded experimental data from each session.

For optogenetic experiments, LED light delivery was controlled through a TTL adapter, converting 28V DC output to a TTL transition (Med Associates, Georgia, Vermont). This was connected to an LED driver (Doric lenses) and Doric Connectorized LED light source with a fibre-optic rotary joint for connection to fibre optic patch cords. For all optogenetic experiments LED light stimulation was delivered as orange light (625 nm) with a power of ~10 mW measured at the cannula tip. Due to the extended time periods of light delivery, and recent reports that continuous light delivery to the striatum is sufficient to heat tissue and suppress spiking (*Owen et al., 2019*), light was pulsed at 40 Hz, which has been shown to reduce tissue heating.

For tests of goal-directed behaviour via sensory-specific satiety, a separate devaluation room was used that contained 16 individual, open top plastic boxes with stainless steel wire mesh lids. During devaluation, lights were kept off and each individual chamber was fitted with either a glass petri dish for pellet devaluation, or a plastic drink bottle with sipper for sucrose devaluation.

## Drugs

For DREADDs experiments, CNO (RTI International) was used as the ligand to activate either the hM4Di or hM3Dq DREADDs receptors. CNO was made on the morning of each required test day, dissolved in 0.6% 5 M HCl and diluted with distilled water (dH$_2$O). Vehicle consisted of 0.1% 5 M HCl diluted with dH$_2$O and pH was matched with that of the CNO in solution. It is generally accepted that CNO is more effective in stimulating activity through hM3Dq DREADDs than it is in inhibiting activity through hM4Di DREADDs and thus, higher doses of CNO are generally used for hM4Di DREADDs in order to induce behavioural effects (see *Campbell and Marchant, 2018* for review). Here we used CNO concentrations that have been shown to induce behavioural effects in rats. For DIO-hM4D, CNO was made to a concentration of 7 mg/ml and injected at a dose of 7 mg/kg (*Hart et al., 2018a*). For DIO-hM3D, CNO was made to a concentration of 3 mg/ml and injected at a dose of 3 mg/kg (*Yau and McNally, 2015*).

## Food restriction

For behavioural experiments, rats underwent 3 days of food restriction before the onset of magazine training. For the first 2 days of food restriction, male and female rats received 8 or 6 g of standard lab chow, respectively, and this increased to 12 or 8 g for the third day and remainder of experiment, adjusted dependent on rats' weight across days. Their weight was monitored closely to ensure that their food restricted body weight did not fall below 85% of their free-feeding body weight.

## Magazine training

Rats were given 2 days of magazine training during which the to-be-trained food rewards (either 45 mg grain pellets and 0.2 ml of 20% sucrose solution or 0.2 ml of 20% polycose solution) were delivered at a random interval schedule of 60 s.

## Instrumental pre-training

For experiments that included pre-training, rats received two daily sessions (one session on each of left and right lever, order counterbalanced within groups and across days) of lever press training each day. Lever press responses were rewarded with 0.2 ml of 20% polycose solution, which on Days 1 and 2 were delivered on a continuous reinforcement schedule (CRF; every lever press earned one reward). On Day 3, lever presses were rewarded on a random interval 15 (RI15; presses were rewarded on average every 15 s). On Days 4 and 5, lever presses were rewarded on a random interval 30 (RI30; presses were rewarded on average every 30 s). Criteria for progression from CRF on to RI15 training was 30 outcomes earned in a single session for each lever.

## Instrumental training

For SPN tracing with immunofluorescence experiment, rats did not receive pre-training and instead received instrumental training sessions following 2 days of magazine training. Rats were trained on either the left or right lever, counterbalanced within groups. Lever press responses were rewarded at increasing intervals over training days. Training sessions progressed from CRF (Day 1) to RI15 (Day 2) to RI30 (Days 3 and 6). *For DREADDs experiments*, following instrumental pre-training rats received 2 days of instrumental training, during which they received alternating presentations of

each lever, which delivered separate outcomes (sucrose solution or grain pellets). Rats were presented with one lever (left or right, counterbalanced within groups), which earned one outcome (either pellets or sucrose, counterbalanced within groups) on an RI30 schedule of reinforcement. This lever was presented for either 10 min (dSPN rats; average 15.1 outcomes) or 15 outcomes (iSPN rats; average 9.2 min), after which it retracted, and the house light went off for a 1 min intertrial interval (ITI), and then the alternate lever was presented with the same criteria. This sequence was repeated, such that all rats received two presentations of each lever. *For optogenetic inhibition experiment,* instrumental training was as described for DREADDs experiments; however, rats received 4 days of instrumental training on an RI30 schedule, with levers presented for 15 outcomes.

## Outcome devaluation
All rats were habituated to devaluation chambers with two 1 hr pre-exposure sessions conducted during the pre-training phase. On test days, rats were placed in devaluation chambers for 1 hr where they had ad libitum access to one of the previously earned outcomes, either sucrose solution or grain pellets. Rats were then immediately returned to the operant chambers for the choice test. Devaluation was conducted across two consecutive days of choice testing (for rewarded and non-rewarded choice tests), with one outcome (pellets or sucrose solution) devalued each day, order counterbalanced within groups.

## Non-rewarded choice test
*For DREADDs experiments,* non-rewarded choice tests were conducted drug free. Sessions began with the illumination of the house light, and simultaneous presentation of both left and right levers. No outcomes were delivered during this session and responses on left and right levers recorded, as well as magazine entries. Sessions ended after 5 min. For *optogenetic inhibition experiment,* non-rewarded choice tests were the same as described for DREADDs experiments, but lasted 7.5 min, during which orange LED light was delivered bilaterally into the brain in an ABA design with LED illumination on for the first and final 2.5 min of testing, separated by 2.5 min of no LED illumination.

## Rewarded choice test
*For DREADDs experiments*, on rewarded choice test days, rats received an injection of CNO or vehicle prior to outcome devaluation and thus approximately 1 hr before test commencement. Both levers were rewarded with their previously earned outcomes (pellets and sucrose), on a random ratio 7 (RR7; on average every 7th press is rewarded) schedule, with the constraint that the first response on each lever was always rewarded. Sessions ended after 10 min. *For optogenetic inhibition experiment,* rewarded choice testing was as described for DREADDs experiments; however, each test lasted 7.5 min, with orange LED light delivered into the brain bilaterally for the first and last 2.5 min.

## Non-rewarded choice test with unilateral LED light delivery
Non-rewarded choice tests were conducted in the absence of devaluation pre-feeding and orange LED light delivered unilaterally into either the left (first test) and right (second test) hemisphere. This was done so that each rat received suppression to both hemispheres separately, and also that for tests 1 and 2, the outcome that was paired with the left or right lever, relative to instrumental training protocol, was counterbalanced within groups. Both tests occurred on the same day, approximately 1 hr apart for each rat. The test itself was identical to that described above for the non-rewarded choice test with bilateral LED light delivery.

## Reversal training
Sessions began with the illumination of the house light, and after 1 min, both the left and right levers were presented. Orange LED light was delivered into the brain for all periods when levers were presented. During lever presentation, only one lever was rewarded on an RI15 schedule of reinforcement (preserving the originally trained response-outcome pairings) and this alternated in an ABBABABAAB design whereby A corresponded to left lever rewarded and B corresponded to right lever rewarded. Each 'block', that is, A or B, lasted for 2.5 min and was separated by a 1-min inter-trial interval, during which both levers were retracted, but the house light remained illuminated. In this test, responding on the left and right levers were recorded during periods of lever presentation,

as well as total magazine entries. Sessions ended after the 10th block (35 min) and rats were returned to their home cages.

## Rotarod

An assessment of general motor coordination was conducted for all DREADDs experiments using the rotarod test. Rats received 1 day of rotarod training, drug free, during which they received three trials on a fixed revolutions per minute (rpm) setting of 5 rpm. Rats then received 2 days of rotarod testing on an accelerating mode, whereby the speed or rotation was set at a minimum of 5 rpm, and this increased to 40 rpm over a 4-min period. On each of these test days rats received an injection of either CNO or vehicle, counterbalanced across rats. On each test day, rats received three trials and the best two trials were averaged to give a value for total time spent on the rotarod in seconds.

## Immunofluorescence procedures

### Transcardial perfusion and tissue sectioning

For tissue analysis, rats received an intraperitoneal injection of a lethal dose of pentobarbital (0.5–0.9 ml) and were perfused transcardially with 400 ml of cold 4% paraformaldehyde (PFA) in 0.1 M phosphate buffer (PB). Fixed brains were immediately removed and stored in PFA for a further 12–72 hr. Brains were sliced into 40 μm coronal sections in 0.1 M PB salt (PBS) solution on a vibratome (LeicaVT1000S) and stored at −30°C in a cryoprotective solution.

### Immunofluorescence protocol

Free-floating sections containing the region of interest were washed three times for 10 min each in 0.1 M PBS solution. Sections were then incubated in a 0.5 Triton-X, 10% NHS in 0.1 M PBS blocking solution for 2 hr. Following this, sections were transferred into a solution containing primary antibodies in 0.2% Triton-X, 2% NHS in 0.1 M PBS for between 12 and 48 hr (dependent upon antibody) at either room temperature or 4°C. Sections were washed three times in 0.1 M PBS for 10 min each and then incubated in a solution containing the secondary antibodies diluted in 0.2% Triton-X, 2% NHS in 0.1 M PBS for 2 hr at room temperature. Sections were again washed three times in 0.1 M PBS for 10 min, and once in 0.1 M PB for 10 min and mounted onto glass slides using Fluoromount or Vectashield mounting medium. Sections that did not require immunofluorescence staining were washed three times for 10 min each in PBS and then for 10 min in PB and mounted in the same fashion. Images were generally taken the day following mounting using a confocal microscope (Olympus BX61WI) or a Spinning Disk Microscope (Andor Diskovery).

### SPN tracing and immunofluorescence experiment

For simultaneous detection of DARPP-32, FG and CTB for tracing analysis, sections containing the pDMS were incubated with mouse anti-DARPP-32 (1:1000) and goat anti-CTB (1:2000). For simultaneous detection of Zif268, FG and CTB for activity analysis, sections containing the pDMS were incubated with rabbit anti-EGR1/Zif268 (1:300) and goat anti-CTB (1:2000). For placement analysis, sections containing the SNr and GPe were selected and immunostained for CTB (1:2000). Native fluorescence was used for identification of FG throughout.

### DREADDs experiments

Sections containing the pDMS were selected and virus expression imaged at the widest point. mCherry expression from DIO-hM4D, DIO-hM3D, and DIO-mCherry were imaged without the addition of fluorescent antibodies. For placement analysis of retro-Cre, SNr sections were incubated with mouse anti-DARPP-32 (1:1000) and rabbit anti-GFP (1:1000) to label SNr boundaries and retro-Cre expression. GPe sections were incubated with rabbit anti-GFP (1:1000).

### Optogenetic experiment

Five sections were selected that contained the pDMS at different anterior–posterior coordinates, and three selected that contained either the SNr (for dSPN targeting) or GPe (for iSPN targeting). pDMS sections were imaged for eYFP without the use of fluorescent antibodies. Verification of SNr and GPe placements of retro-Cre was the same as described for DREADDs experiments.

All immunofluorescence staining used secondary antibodies Alexa 488, Alexa 546, or Alexa 647 (1:1000).

## Electrophysiology

### Brain slice preparation

Rats were euthanized under deep anaesthesia (isoflurane 4% in air). Brains were rapidly removed and cut using a vibratome (Leica Microsystems VT1200S, Germany) in ice-cold oxygenated sucrose buffer containing (in mM): 241 sucrose, 28 $NaHCO_3$, 11 glucose, 1.4 $NaH_2PO_4$, 3.3 KCl, 0.2 $CaCl_2$, and 7 $MgCl_2$. Coronal brain slices (300 μm thick) containing the pDMS were sampled and maintained at 33°C in a submerged chamber containing physiological saline with composition (in mM): 124 NaCl, 3.5 KCl, 1.25 $NaH_2PO_4$, 1 $MgCl_2$, 1 $CaCl_2$, 10 glucose, and 26 $NaHCO_3$, and equilibrated with 95% $O_2$% and 5% $CO_2$.

### Electrophysiology recording

After equilibration for 1 hr, slices were transferred to a recording chamber and neurons visualized under an upright microscope (BX50WI, Olympus, Shinjuku, Japan) using differential interference contrast (DIC) Dodt tube optics and eYFP/mCherry fluorescence, and superfused continuously (1.5 ml/min) with oxygenated physiological saline at 33°C. Whole-cell patch-clamp recordings were made using electrodes (2–5 MΩ) containing internal solution (in mM): 115 K gluconate, 20 NaCl, 1 $MgCl_2$, 10 HEPES, 11 EGTA, 5 Mg-ATP, and 0.33 Na-GTP, pH 7.3, osmolarity 285–290 mOsm/l. Biocytin (0.1%) was added to the internal solution for marking the sampled neurons during recording. Data acquisition was performed with a Multiclamp 700B amplifier (Molecular Devices, Sunnyvale, CA), connected to a Macintosh computer and interface ITC-18 (Instrutech, Long Island, NY). Liquid junction potentials of −10 mV were not corrected. In current-clamp mode, membrane potentials were sampled at 5 kHz (low pass filter 2 kHz, Axograph X, Axograph, Berkeley, CA). Stock solution of drug was diluted to working concentration in the extracellular solution immediately before use and applied by continuous superfusion. Data from whole-cell recordings were only included in analyses if (1) the neurons appeared healthy under DIC on monitor screen and (2) action potential amplitudes were at least 65 mV measured under current-clamp mode, to ensure that only highly viable neurons were included.

### LED stimulation

For light-evoked terminal glutamate release impinged onto SPNs, ChR2-containing corticostriatal axons in the pDMS were stimulated by pulsing an LED light (473 nm, ThorLabs, New Jersey, LED light source attached to microscope) onto the slice under 40× water-immersion objective. Continuous trains (at 0.5 Hz) of light pulses (1 ms width, interpulse interval 50 ms and 20 pulses per train) were delivered onto the slice at a maximum intensity of 3.9 mW to evoke action potentials arising from excitatory postsynaptic potentials. The stimulation protocol was chosen to mimic in vivo recordings in SPNs. For halorhodopsin experiment, EYFP positive and negative SPNs were injected repeatedly with +25, +50, or +100 pA current (duration 200 ms pulse, triggered at 0.5 Hz) to evoke action potentials. After initial 10 s baseline, LED 625 nm (1 mW) was shone onto the slice pulsing at 40 Hz for 150 s, to mimic our in vivo experimental protocol.

### Post hoc histological analysis

Immediately after physiological recording, brain slices containing biocytin-filled neurons were fixed overnight in 4% paraformaldehyde/0.16 M PB solution, rinsed, and then placed in 0.3% Triton X-100/PB for 3 days to permeabilize cells. Slices were then incubated in AMCA-conjugated avidin (1:500), or streptavidin-conjugated Alexa 488 (1:500), plus 2% horse serum and 0.2% Triton X-100/PB for 2 hr to reveal biocytin-labelled neurons. Halorhodopsin-EYFP immunofluorescence signal was enhanced with a rabbit anti-GFP antibody (1:1000 for 2 days) followed by a secondary donkey anti-rabbit Alexa 488 conjugated IgG (1:1000 overnight). Stained slices were rinsed, mounted onto glass slides, dried, and coverslipped with Fluoromount-G mounting medium (Southern Biotech). A 2D projection Image was later obtained from a collated image stack using confocal laser scanning microscopy (Fluoview FV1000, BX61WI microscope, Olympus).

## Analyses

### Exclusions and group allocation

SPN tracing and immunofluorescence. Three rats were excluded from analyses; two for misplaced injection sites leaving 23 rats and one due to damaged sections that were not quantifiable. This left 22 rats for analysis (Group Instrumental, n=11; Group Yoked, n=11).

### DREADDs suppression experiment

Twenty-eight rats were excluded; 15 for low virus expression (<25 cells/mm$^2$), 8 for virus spread beyond the DMS (to either the GPe or DLS), 1 for damage, 3 that did not acquire the instrumental response, and 1 that did not consume the outcome during devaluation pre-feeding. This left 51 rats for analysis, of which 23 were males and 28 were females (Group CNO control, n=13; Group dSPN+-VEH, n=16; Group iSPN+VEH, n=4; dSPN+CNO, n=12; Group iSPN+CNO, n=6).

### DREADDs stimulation experiment

Ten rats were excluded; five for low virus expression in the pDMS (<25 cells/mm$^2$) and five for excessive virus spread not confined to the pDMS, leaving 20 rats for analysis, of which 11 were males and 9 were females (Group mCherry+CNO, n=6; Group hM4D+VEH, n=6; Group hM4D+CNO, n=8).

### Optogenetic inhibition experiment

Fourteen rats were excluded; 2 for low virus expression in the pDMS, eleven for misplaced cannulae, and 1 for damage, leaving 25 rats, of which 11 were males and 14 were females (Group eYFP control, n=11; Group dSPN eNpHR, n=6; Group iSPN eNpHR, n=8).

### Immunofluorescence analysis

All immunofluorescence analysis was conducted using Image J software. Animals with misplaced injection sites or misplaced cannulae were excluded. Statistical analyses were performed using PSY software. The per-comparison error rate was controlled at α=0.05.

### Activity marker

To maintain consistency for Zif268 quantification, all sections within a series were imaged within 3 days of immunofluorescence staining, and all images were taken within a 24 hr period. A brightness threshold was set for each series of sections that were immunostained and imaged together, and this was kept constant within a series. Only cells containing two or more pixels brighter than these values were counted. Cell counts were analysed according to the total number of cells, calculated as the mean number of cells per mm$^2$, per hemisphere, averaged across four slices per rat. Mean Zif268 was also calculated as a percentage of total dSPN or iSPN pathway; cells that were positive for Zif268 and either FG or CTB were expressed as a percentage of the total number of FG or CTB positive cells. Co-labelled cells, that is, Zif268+FG or Zif268+CTB do not include counts for triple labelled cells, that is, Zif268+FG+CTB. Orthogonal contrasts were used to test for a main effect of group and orthogonal pairwise comparisons between each hemisphere in each group.

### Retrograde tracing

For tracing analysis, cells that expressed FG, CTB, and DARPP-32, as well as all co-labelled cells, were quantified and averaged across hemispheres, to give a mean number per mm$^2$ for each hemisphere for each rat. Cells that were positive for both DARPP-32 and FG or CTB were represented as a percentage of total DARPP-32 positive neurons. Co-labelled cells, that is, DARPP-32+CTB or DARPP-32+FG do not include counts for triple labelled cells, that is, DARPP-32+FG+CTB. Pairwise comparisons were used to test for differences in how many neurons were labelled with each retrograde tracer, FG or CTB and orthogonal contrasts were used to test for differences in the number of neurons that were labelled with each tracer in the pDMS in instrumental versus yoked rats.

### DREADDs expression

Counts of mCherry positive cells in the pDMS were conducted manually with a cell counter in Image J and total number of cells expressed as cells/mm$^2$. Orthogonal contrasts were used to compare the

number of cells expressing DIO-hM4D between groups that expressed this virus on the same population of SPNs, that is, dSPNs or iSPNs and to compare the number of cells infected with each virus (DIO-hM4D or DIO-mCherry).

Data from electrophysiological recordings were analysed using PSY software. Action potential frequency was measured at three different, 1 s time points: 1 min prior to CNO application, immediately prior to CNO application, and 1 min after CNO application. Pairwise comparisons were used to compare the action potential frequency between the two time points prior to CNO application, as well as to compare the action potential frequency after CNO application to the time point immediately before CNO application.

### Behavioural analysis

For behavioural experiments, data collection was performed using Med-PC software (Versions IV and V), which was tabulated using Microsoft Excel and graphs generated with GraphPad Prism 8. Statistical analyses were performed using PSY software. The per-comparison error rate was controlled at $\alpha$=0.05. For non-orthogonal contrasts, a Bonferroni correction was used to control the family-wise error rate at $\alpha$=0.05.

### SPN activity marker experiment

Training data was analysed using orthogonal contrasts, testing for a main effect of group (instrumental versus yoked) and a linear trend analysis used to examine changes in magazine entries and press rates across training days.

### DREADDs inhibition experiment

Non-orthogonal contrasts were used for all data analyses, and the family-wise error rate (FWER) controlled at $\alpha$=0.05 using Bonferroni correction. All analyses tested for a main effect of group (dSPN +CNO versus controls, iSPN+CNO versuscontrols, and vehicle controls versus CNO control). For instrumental training, data was analysed for a main effect of Group and training day, and any interactions. Non-rewarded choice test data was analysed for a main effect of Group and lever (devalued versus valued) and any interactions (Group $\times$ Lever). Rewarded choice test data was analysed for a main effect of Group and lever (devalued versus valued) and any interactions (Group $\times$ Lever). Rotarod data was analysed for a pairwise comparisons within each group, when rats were tested under CNO or vehicle.

### DREADDs stimulation experiment

Planned orthogonal contrasts were used for all data analyses. Training data was analysed testing for a main effect of group (hM3D+CNO versus controls, hM3D+VEH versus CNO control) and a main effect of training day. Non-rewarded choice test data was analysed testing for a main effect of group (as above) and a main effect of lever (devalued versusvalued).

### Optogenetic inhibition experiment

Training data were analysed using orthogonal contrasts, testing for a main effect of group (eNpHR groups versus controls, dSPN eNpHR versus iSPN eNpHR) on rates of responding across days. Test data were analysed using the same contrasts, testing for effects of Group, LED (LED ON or LED OFF), and lever (devalued versus valued, or ipsi versus contra). Rotational behaviour was analysed testing for the effect of LED on ipsilateral and contralateral rotations in each group. Reversal training data was analysed using orthogonal contrasts testing for effects of Group (eNpHR groups versus controls, dSPN eNpHR versus iSPN eNpHR) on pressing the rewarded and non-rewarded lever and their difference score on each day. Follow-up analysis of the same data was performed using non-orthogonal contrasts controlling the FWER at $\alpha$=0.05 using Bonferroni correction, comparing Group dSPN eNpHR to Group eYFP control and Group iSPN eNpHR to Group eYFP control on each day. Finally, trial-by-trial reversal data were analysed separately across each day with non-orthogonal contrasts testing for a main effect of reversal (lever $\times$ trial type interaction) and reversal $\times$ group interactions (Groups iSPN and dPSN each against control) within each day, controlling the FWER at $\alpha$=0.05 using Bonferroni correction.

## Acknowledgements

The research reported in this manuscript was supported by a Discovery Grant from the Australian Research Council, #DP150104878, and both a Project Grant, #GNT 1165346, and a Senior Principal Research Fellowship, #GNT1079561, from the National Health and Medical Research Council of Australia to BWB. The authors thank J Bertran-Gonzalez for comments on the manuscript.

## Additional information

### Funding

| Funder | Grant reference number | Author |
|---|---|---|
| Australian Research Council | DP150104878 | Bernard W Balleine |
| National Health and Medical Research Council | GNT1165346 | Bernard W Balleine |
| National Health and Medical Research Council | GNT1079561 | Bernard W Balleine |

The funders had no role in study design, data collection and interpretation, or the decision to submit the work for publication.

### Author contributions

James Peak, Billy Chieng, Conceptualization, Data curation, Formal analysis, Investigation, Methodology; Genevra Hart, Conceptualization, Formal analysis, Investigation, Methodology, Project administration; Bernard W Balleine, Conceptualization, Funding acquisition, Project administration

### Author ORCIDs

Genevra Hart (iD) https://orcid.org/0000-0002-8463-9869
Bernard W Balleine (iD) https://orcid.org/0000-0001-8618-7950

### Ethics

Animal experimentation: All experiments conformed to the guidelines on the ethical use of animals maintained by the Australian code for the care and use of animals for scientific purposes, and all procedures were approved by the Animal Care and Ethics Committee at either the University of New South Wales (Protocol number 19/25A) or the University of Sydney (Protocol number 5960/78). All surgery was performed under isofluorane anesthesia, and every effort was made to minimize suffering.

### Decision letter and Author response

Decision letter https://doi.org/10.7554/eLife.58544.sa1
Author response https://doi.org/10.7554/eLife.58544.sa2

## Additional files

### Supplementary files

• Transparent reporting form

### Data availability

All data generated or analysed during this study are included in the manuscript and supporting files. Source data files have been provided for all experiments reported in this manuscript in an online repository at https://figshare.com/s/23578523b81df00fa6e4.

The following dataset was generated:

| Author(s) | Year | Dataset title | Dataset URL | Database and Identifier |
|---|---|---|---|---|
| Peak J, Chieng B, Hart G, Balleine BW | 2020 | Striatal direct and indirect pathway neurons differentially control the encoding and updating of goal-directed learning | https://figshare.com/s/23578523b81df00fa6e4 | figshare, 23578523b81df00fa6e4 |

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
