## [Decision Letter]

**Acceptance summary:**

Animals adapt to changing environments to maximize opportunities for reward by adjusting their actions to meet their goals. This study uncovers a role for one distinct projection neuron subtype involved in learning of goal directed actions, while a second neuron population is necessary for adjusting these behaviors. Overall, this study provides new insights into the specific neuron populations, in the complex brain, that are important for behaviors involved in adapting to new environments.

**Decision letter after peer review:**

[Editors’ note: the authors submitted for reconsideration following the decision after peer review. What follows is the decision letter after the first round of review.]

Thank you for submitting your work entitled "Striatal direct and indirect pathway neurons differentially control the encoding and updating of goal-directed learning" for consideration by eLife. Your article has been reviewed by a Senior Editor, a Reviewing Editor, and three reviewers. The reviewers have opted to remain anonymous.

Our decision has been reached after consultation between the reviewers. Based on these discussions and the individual reviews below, we regret to inform you that your work will not be considered further for publication in eLife.

The reviewers considered the examination of the role of the direct and indirect pathway from the dorsomedial striatum on learning and expression of goal-directed action to be of importance. They were positive about both the circuit specific complementary approaches and the use of rats since previous work has mainly used mouse lines. While the reviewers considered that the work could be impactful the major concerns about statistical analysis, the imbalance of animal numbers between dSPN and iSPN groups, and the lack of validation of the wavelength used in the eNpHR optogenetic studies weakened the conclusions of the study. If the authors are able to address these major concerns including the statistical analysis, which could include a statistician consult to support the design and analysis used in the current study, and address additional concerns listed in the reviews they are encouraged to submit a new version of the manuscript to eLife.

Reviewer #1:

This was a well written paper examining the contribution of the direct and indirect pathway to the learning and expression of goal-directed action control in the pDMS of the dorsal striatum. The area has long been implicated in controlling the acquisition and expression, but the functional contributions to these processes of the two types of SPNs in the striatum has not been clear. Here, the authors take a projection specific approach, using retrograde Cre targeted to downstream projection areas of the two populations, with functional (DREADD and Halo) manipulation approaches. The use of projections to target (instead of genetically defined) may because of the use of rats, but the authors highlight the collateral issue (although the projection approach found fairly high collateral overlap as well). Rats were trained on an instrumental task and they found that EGR1 was upregulated significantly in dSPNs, but not iSPNs. To follow this up, they used chemogenetics to suppress activity of dMSNs during training and saw deficits in goal-directed behavior as assessed through an outcome devaluation test, without seeing this effect in inhibited iSPNs. They went on to further characterize the moment-to-moment influence of dSPNs on behavior using halorhodopsin to silence neurons. They found that stimulation biased an ipsilateral response. Finally, they inhibited iSPNs and saw some amount of delay or deficit in the acquisition of reversal learning. While the indirect pathway inhibition revealed a proposed role for response flexibility, but there the data may be a bit weaker. These findings support evolving ideas about how the interplay between D1 and D2 SPNs support striatal computations. Overall, the paper was well written, more details on methods should be included in the script, as many readers may not be as well versed in these tasks, and the statistical reporting was largely fine. More comments below.

- A little worried that the halorhodpsin experiments are not conclusive. One concern is the use of the 625nm wavelength LED, which is on the lower tail of its excitation wavelength range, was not sufficient to drive effect channel function and inhibit the cell. Further, there is no slice or in vivo validation of inhibition with that LED wavelength for the times used, something the lab is easily capable of doing as shown in the ChR2 experiments.

- Looking at the lack of bilateral inhibition, it is hard to make a conclusion from this. That there is a bias to the contralateral side with ipsilateral inhibition is somewhat reassuring, but again, it is not clear how effective inhibition was. Clarification of that would go a long way in raising confidence in these results.

- In addition, how to analyze Figure 5F-H is tricky. I understand the stats the authors did, and that seems fine for one approach. But looking at the data patterns, it is hard to conclude that only when inhibiting iSPN is there an effect. The dSPN and iSPN groups look very similar. First, it appears that iSPN and dSPN rats show a LL+ bias, that is absent in the control rats (first lever encountered). Within genotype, is there a main effect of lever? The concern is overstating the isolated effect in the indirect pathway. Perhaps better layout of the stats (main effects) and comparison between the two experimental groups would help shed light on the seeming similarity between the groups.

Reviewer #2:

In this paper, Peak et al., examine the role of the posterior dorsomedial striatum (pDMS) in goal-directed action. They show that direct pathway neurons (dSPNs) in the pDMS were involved in instrumental acquisition and indirect pathway neurons (iSPNs) are important for refining that learning. These experiments were impressively conducted in rats, which is welcome as most studies of these two striatal populations have been done in mice. Despite this strength, the statistical comparisons were difficult to follow and appeared to have flaws throughout. Other interpretations for their results are also not considered. Both significantly reduced my enthusiasm for this paper.

Essential revisions:

- I had concerns about how the ANOVAs were set up throughout. Mostly, these concerns came from examination of the DFs, which do not appear to reflect the experimental design for almost all figures. For instance, from their degrees of freedom for Figure 2 it appears to be a one-way ANOVAs comparing between 2 conditions, as they report DF (1,46) throughout all results. However, they have 5 groups and 2 conditions per group. I'm not sure how these ANOVAs were set up to evaluate this data set and come to DF (1, 46). This issue comes up again in other figures. Figure 4H shows data from 3 groups, 2 illumination conditions, and two sides, and again reports df1 = 1. Were all groups combined into 2 groups and then tested on illumination or side independently? In figure 4L, they report DF(1,5), suggesting they may have run the ANOVA on *means* of the derived values in Figure 4K for each group, resulting in 6 numbers from 2 groups in this analysis? I am not sure though, as the figure shows error bars suggesting that each condition had multiple data points in it. Figure 5 has 3 groups, 2 levers, and 2 tasks states, and again reports df1 = 1 for all ANOVA results. This was concerning throughout, and left me confused about what the statistical results meant.

- Different data presentation formats were used throughout, giving the feeling that different people prepared different figures. While this is not inherently problematic, I did not like how in some graphs individual data points were shown while others only presented group means and error bars. Especially given the potential issues with ANOVA reporting this inhibited my ability to evaluate the data as presented. A consistent format showing individual data points should be presented throughout.

- The authors conclude that that goal-directed learning is dependent upon dSPNs, as devaluation was not apparent after inhibiting dSPNs during instrumental learning. They define this test for goal-directed action as, "Goal-directed action is demonstrated by rats showing a reduction in responding on the lever previously associated with the now-devalued outcome relative to the other lever". However, they don't show the levels of lever-pressing before devaluation, so we can't evaluate if the responding on the devalued lever changed. Rather, they infer a deficit in goal-directed action by a lack of difference in responding on the lever previously associated with the devalued outcome, relative to the non-devalued lever. This inference requires an assumption that all groups had similar responding on all levers before devaluation, which they never show. I would have like to see the devaluation test as described in their definition.

My issue here is that without seeing the baseline responding on each lever, it is ambiguous whether changes are specific to the devalued lever or whether there may be a more general change in behavior towards both levers. This could indicate a change in overall motivation for the outcomes. In the data in Figure 2K, relative to the control groups it does not appear that there is heightened responding on the devalued lever in the dSPN-CNO group. Rather, there is a reduction in responding on the lever paired with the *valued* outcome relative to the control group (dSPN+CNO vs dSPN+VEH). This experiment leaves open the possibility that inhibiting dSPNs impaired their overall motivation for the outcome. As such I found it difficult to support their conclusion that the difference in Figure 2K was due solely to a deficit in goal-directed learning.

Finally, the number of animals in each group for this experiment was highly variable. Only 4 animals were in the iSPN + veh group, and 6 in the iSPN + CNO group. There were much larger numbers of dSPN animals in this experiment (by my count, n=15 dSPN + veh and n=12 in the dSPN + CNO group). As the conclusion of this experiment was that dSPNs were involved in goal directed action, it seems the experiment was more strongly powered to find an effect in dSPNs and not iSPNs. No explanation was given for the large discrepancy in animals of each group. I obtained these Ns from counting dots on Figure 2K, it would be nice to report Ns in the text or figure captions.

- In Figure 4, they map unilateral manipulations of the striatal pathways to responding on the right or left side of the chamber. However, while the levers are on different sides of the chamber, the animals are freely moving and can press either lever however they want, with either contralateral or ipsilateral limbs or directional actions. So, despite the lever position in the chamber, it is not clear if the lever-pressing is in fact ipsilateral or contralateral to the stimulation side. The effect also appears to be driven most strongly by a side bias in the dSPN eNPHR group when the stimulation is off.

- No discussion is made on potential motor or rotational effects of stimulating these pathways in their optogenetic experiments, which was quite strong (10mW) and high frequency (40Hz). Changes in performance may be due to non-specific rotational behavior disrupting the animal's ability to perform these tasks, which was not measured or discussed.

- There were some scholarship issues with citations being incorrectly attributed. For instance, the manuscript incorrectly cites Kim et al., 2009 with the sentence "bilateral stimulation of dSPNs tends to increase, and of iSPNs to retard, both spontaneous movement and the flexible performance of reward-related actions". There was no stimulation or cell type specific experiments in this paper. In this same sentence they incorrectly cite Hikida et al., 2010, which did not use stimulation but reversible synaptic transmission blockade of each pathway, and Kwak and Jung, 2019, which also used inhibition and did not report on motor actions of stimulating these pathways. I didn't exhaustively check every cited reference but these jumped out at me as I knew the papers well. The authors should check that all references are correctly attributed.

Reviewer #3:

In this study, the authors examined the role of the dorsomedial striatum in the encoding and updating of goal-directed behaviour. Using a set of circuit specific complementary approaches, the authors found that learning of goal-directed behaviour recruits the medium size spiny neurons of the direct pathway (dSPN) whereas updating of this learning specifically recruits the medium size spiny neurons of the indirect pathway (iSPN). The topic investigated by the authors is important. The methods are appropriate and elegant and the experiments very well executed and analysed. I think that the results significantly add to the field. My comments are listed below:

1) The authors included both male and female animals. Did they find any indication of a gender effect?

2) Past research has demonstrated that the anterior and posterior portions of the DMS are differentially involved in the control of goal directed learning and performance. As a result, the figures showing virus infections should include the anatomical boundaries of the infections along the antero-posterior axis.

3) The design of Experiment 2 (see Figure 2) with A1-O3, A2-O3 followed by A1-O1, A2-O2 might create some sort of outcome identity reversal learning. Could the authors comment on the absence of iSPN manipulation?

4) Previous research of the same group has provided evidence for a role of cholinergic striatal interneurons (CIN) in the updating of goal-directed behaviour. Could the authors include a discussion about the possible relative contribution of CIN and iSPN neurons in this process?

---

## [Author Response]

[Editors’ note: The authors appealed the original decision. What follows is the authors’ response to the first round of review.]

Reviewer #1:This was a well written paper examining the contribution of the direct and indirect pathway to the learning and expression of goal-directed action control in the pDMS of the dorsal striatum. The area has long been implicated in controlling the acquisition and expression, but the functional contributions to these processes of the two types of SPNs in the striatum has not been clear. Here, the authors take a projection specific approach, using retrograde Cre targeted to downstream projection areas of the two populations, with functional (DREADD and Halo) manipulation approaches. The use of projections to target (instead of genetically defined) may because of the use of rats, but the authors highlight the collateral issue (although the projection approach found fairly high collateral overlap as well). Rats were trained on an instrumental task and they found that EGR1 was upregulated significantly in dSPNs, but not iSPNs. To follow this up, they used chemogenetics to suppress activity of dMSNs during training and saw deficits in goal-directed behavior as assessed through an outcome devaluation test, without seeing this effect in inhibited iSPNs. They went on to further characterize the moment-to-moment influence of dSPNs on behavior using halorhodopsin to silence neurons. They found that stimulation biased an ipsilateral response. Finally, they inhibited iSPNs and saw some amount of delay or deficit in the acquisition of reversal learning. While the indirect pathway inhibition revealed a proposed role for response flexibility, but there the data may be a bit weaker. These findings support evolving ideas about how the interplay between D1 and D2 SPNs support striatal computations. Overall, the paper was well written, more details on methods should be included in the script, as many readers may not be as well versed in these tasks, and the statistical reporting was largely fine. More comments below.- A little worried that the halorhodpsin experiments are not conclusive. One concern is the use of the 625nm wavelength LED, which is on the lower tail of its excitation wavelength range, was not sufficient to drive effect channel function and inhibit the cell. Further, there is no slice or in vivo validation of inhibition with that LED wavelength for the times used, something the lab is easily capable of doing as shown in the ChR2 experiments.

We agree with the reviewer’s concerns and have now conducted a full ex vivo validation study of the halorhodopsin parameters used in our experiments. Using the same surgical methods described in our original submission, rats were injected with retro-Cre into either the SNr or GPe and Cre dependent halorhodopsin was injected into the pDMS. Action potentials were evoked with an injection of +50 or +100 pA current (duration 200 ms pulse, triggered at 0.5 Hz) to evoke action potentials. The delivery of a 625nm wavelength LED (pulsing at 40Hz for 150 seconds, 1mW) was effective in inhibiting firing, relative to baseline (See Figure 4A-C and Figure 4—figure supplement 1C). As is evident in the figure, firing was clearly and continuously suppressed by LED light delivery and returned to baseline levels following the cessation of that LED light delivery. In addition to these findings, we recorded from SPNs that did not express eYFP (DIO-eNpHR) in the same animals and found no light-induced changes in firing rate when LED light was delivered to the slice (See Figure 4—figure supplement 2A, B and D). We have added a description of this experiment with accompanying analyses in the results section and the description of the methodology in the Materials and methods section.

- Looking at the lack of bilateral inhibition, it is hard to make a conclusion from this. That there is a bias to the contralateral side with ipsilateral inhibition is somewhat reassuring, but again, it is not clear how effective inhibition was. Clarification of that would go a long way in raising confidence in these results.

We appreciate the reviewer’s point here. To clarify, unilateral inhibition of dSPNs induced a bias in responding on the lever ipsilateral to the hemisphere that received optogenetic inhibition. As we describe above, we now provide a study validating the parameters used in our behavioural experiments using ex vivo slice electrophysiology to demonstrate the effectiveness of halorhodopsin inhibition of SPNs using 625nm wavelength LED.

- In addition, how to analyze Figure 5F-H is tricky. I understand the stats the authors did, and that seems fine for one approach. But looking at the data patterns, it is hard to conclude that only when inhibiting iSPN is there an effect. The dSPN and iSPN groups look very similar. First, it appears that iSPN and dSPN rats show a LL+ bias, that is absent in the control rats (first lever encountered). Within genotype, is there a main effect of lever? The concern is overstating the isolated effect in the indirect pathway. Perhaps better layout of the stats (main effects) and comparison between the two experimental groups would help shed light on the seeming similarity between the groups.

The reviewer is correct that there is a small left lever bias in the eNpHR groups, although these mice were genetically identical to the eYFP group and so the bias isn’t explained by genotype effects. However, to address these comments, we have amended the analyses for Figure 4 and Figure 5 such that the main effects are now assessed according to orthogonal contrasts comparing the control group to the two inhibition groups, and the two inhibition groups to each other. This analysis confirms that the iSPN group show a significant impairment relative to the dSPN group during reversal on Day 1, and that this effect is marginal on Day 2. These comparisons were followed up by direct between-group comparisons of the control group vs. each virus group (using Bonferroni correction). As previously, the individual day-by-day data were analysed separately with non-orthogonal contrasts used to test for a main effect of reversal (lever x trial type interaction) and for reversal x group interactions within each day (again using Bonferroni correction). The results of these analyses are described in subsection “Optogenetic inhibition of iSPNs, but not dSPNs, in the pDMS impairs flexible action selection in response to changes in the action-outcome contingency”.

Reviewer #2:In this paper, Peak et al., examine the role of the posterior dorsomedial striatum (pDMS) in goal-directed action. They show that direct pathway neurons (dSPNs) in the pDMS were involved in instrumental acquisition and indirect pathway neurons (iSPNs) are important for refining that learning. These experiments were impressively conducted in rats, which is welcome as most studies of these two striatal populations have been done in mice. Despite this strength, the statistical comparisons were difficult to follow and appeared to have flaws throughout. Other interpretations for their results are also not considered. Both significantly reduced my enthusiasm for this paper.Essential revisions:- I had concerns about how the ANOVAs were set up throughout. Mostly, these concerns came from examination of the DFs, which do not appear to reflect the experimental design for almost all figures. For instance, from their degrees of freedom for Figure 2 it appears to be a one-way ANOVAs comparing between 2 conditions, as they report DF (1,46) throughout all results. However, they have 5 groups and 2 conditions per group. I'm not sure how these ANOVAs were set up to evaluate this data set and come to DF (1, 46). This issue comes up again in other figures. Figure 4H shows data from 3 groups, 2 illumination conditions, and two sides, and again reports df1 = 1. Were all groups combined into 2 groups and then tested on illumination or side independently? In figure 4L, they report DF(1,5), suggesting they may have run the ANOVA on *means* of the derived values in Figure 4K for each group, resulting in 6 numbers from 2 groups in this analysis? I am not sure though, as the figure shows error bars suggesting that each condition had multiple data points in it. Figure 5 has 3 groups, 2 levers, and 2 tasks states, and again reports df1 = 1 for all ANOVA results. This was concerning throughout, and left me confused about what the statistical results meant.

We appreciate the reviewer’s careful analysis of the statistics and our comments here are made with aim of clarifying our statistical approach. The main point of confusion here seems to be driven by the reviewer’s assumption that we used ANOVA for the primary analyses in all experiments. As noted in subsection “Analyses”, behavioural data were analysed using either orthogonal contrasts, controlling the PCER at alpha=0.05 or in the case of non-orthogonal contrasts, Bonferroni was applied to control the FWER at alpha=0.05. This is a widely used and well-validated statistical approach that allows controlled, directional comparisons between groups, and is preferable to ANOVA in experimental cases in which group designs do not neatly fit into a 2 x 2 factorial or comparisons between two means (see Hays, (1973)). For example, in our experiment using DREADDs inactivation, that used 3 control groups and 2 experimental groups, a significant omnibus ANOVA would be meaningless; it could, for example, reflect a significant difference between the control groups. Instead, we have planned logical contrasts that directly compare the control groups, and then the experimental groups and the controls. In these cases, the F-statistic used is not the same as an ANOVA F-statistic, and instead applies F_1, n2_ with n_2_ = N-J. Author response table 1 shows the resulting df applied for each experiment. The precise contrasts tested are described in subsection “Analyses’”.

**Author response table 1. resptable1:** 

Experiment #	N (total rats)	J (number of groups)	df
SPN tracing and immunofluorescence	22	2 (inst vs yoked)	20
M4 DREADDs	51	5 (CNO control, dSPN+VEH, iSPN+VEH, dSPN+CNO, iSPN+CNO)	46 (subset of rats analysed separately N=22, J=4 (no mCherry), df=18)
M3 DREADDs	20	3 (mCherry+CNO, hM4D+VEH, hM4D+CNO)	17
Optogenetic inhibition	25	3 (eYFP control, dSPN eNpHR, iSPN eNpHR)	22

The exception to this was in the final experiment with halorhodopsin inhibition. Here, the training and test data were initially analysed using a two-way, repeated measures ANOVA, followed up by tests for simple effects within each group separately (giving rise to df 5 for a group of n=6). In this revised version of the manuscript, however, we have amended this analysis following the comments of reviewer 1. Taking the analyses of the other experiments together with the comments of reviewer 1 requesting comparisons between iSPN and dSPN inhibition groups, test data from the final study are now also analysed using orthogonal contrasts, comparing the controls to the two inhibition groups and the two inhibition groups to each other. Followed up with non-orthogonal contrasts employing Bonferroni correction.

- Different data presentation formats were used throughout, giving the feeling that different people prepared different figures. While this is not inherently problematic, I did not like how in some graphs individual data points were shown while others only presented group means and error bars. Especially given the potential issues with ANOVA reporting this inhibited my ability to evaluate the data as presented. A consistent format showing individual data points should be presented throughout.

We agree with the reviewer and have now included individual data points throughout all figures for the relevant graphs.

- The authors conclude that that goal-directed learning is dependent upon dSPNs, as devaluation was not apparent after inhibiting dSPNs during instrumental learning. They define this test for goal-directed action as, "Goal-directed action is demonstrated by rats showing a reduction in responding on the lever previously associated with the now-devalued outcome relative to the other lever". However, they don't show the levels of lever-pressing before devaluation, so we can't evaluate if the responding on the devalued lever changed. Rather, they infer a deficit in goal-directed action by a lack of difference in responding on the lever previously associated with the devalued outcome, relative to the non-devalued lever. This inference requires an assumption that all groups had similar responding on all levers before devaluation, which they never show. I would have like to see the devaluation test as described in their definition.

All devaluation tests were conducted twice, once with each lever as the devalued lever whereby the identity of the devalued outcome and lever were counterbalanced within and between groups. The data presented are the average of the two tests. This allows us to ensure that devaluation is a within-subjects factor and for the baselines for each action to contribute equally to both devalued and non-devalued performance. This was described in the methods, but we have now emphasized this point in the Results section too.

My issue here is that without seeing the baseline responding on each lever, it is ambiguous whether changes are specific to the devalued lever or whether there may be a more general change in behavior towards both levers. This could indicate a change in overall motivation for the outcomes. In the data in Figure 2K, relative to the control groups it does not appear that there is heightened responding on the devalued lever in the dSPN-CNO group. Rather, there is a reduction in responding on the lever paired with the *valued* outcome relative to the control group (dSPN+CNO vs dSPN+VEH). This experiment leaves open the possibility that inhibiting dSPNs impaired their overall motivation for the outcome. As such I found it difficult to support their conclusion that the difference in Figure 2K was due solely to a deficit in goal-directed learning.

We understand the reviewer’s point and we have taken steps to address this kind of issue. First, it is important to note that the test data described here (Figure 2K) were collected in a test conducted drug-free. As such, all circuits were intact at the time of test. An explanation along the lines suggested (i.e. that inhibiting dSPNs impairs overall motivation for the outcome) would be indicated by a change in the performance of this group (dSPN + CNO) during training, when neurons were inhibited, of which there was no evidence; as stated in the results, there were no effects of group on baseline responding during the choice extinction test, which goes against the claim that a specific group (i.e., dSPN+CNO) showed general reduction in their motivation to respond. This explanation is also assessed directly in the rewarded choice test conducted under CNO (Figure 4—figure supplement 2H). In this test the outcomes are delivered as in training and it is clear that, in this situation, rats with inhibition of dSPNs were able to show intact preference for the valued over the devalued outcome, indicating that the impairment observed in Figure 2K was specific to a deficit in goal-directed learning during training.

Regarding the reduction in responding on the valued lever, rather than an increase in responding on the devalued lever; this is a commonly observed deficit in outcome devaluation after relatively limited training. There are theoretical reasons why this is likely, including that the “goal-directed” component of responding is that primarily driving the heightened performance on the valued lever, meaning that any loss of that goal-directed control should be most clearly observed (between groups) in responding on that lever. However, it is important to recognise that, in choice tests of this kind, it is the difference in performance between valued and devalued levers that is of most importance. Because responses are made in a choice situation with both levers available simultaneously, comparisons of absolute response rates on one or another lever are problematic; rats showing devaluation do not switch between levers as often as those responding indifferently, imposing a change-over cost in the latter group.

- Finally, the number of animals in each group for this experiment was highly variable. Only 4 animals were in the iSPN + veh group, and 6 in the iSPN + CNO group. There were much larger numbers of dSPN animals in this experiment (by my count, n=15 dSPN + veh and n=12 in the dSPN + CNO group). As the conclusion of this experiment was that dSPNs were involved in goal directed action, it seems the experiment was more strongly powered to find an effect in dSPNs and not iSPNs. No explanation was given for the large discrepancy in animals of each group. I obtained these Ns from counting dots on Figure 2K, it would be nice to report Ns in the text or figure captions.

Group N’s are reported in subsection “Analyses” which were CNO control n=13; dSPN+VEH n=16; iSPN+VEH n=4; dSPN+CNO n=12; iSPN+CNO n=6. The large discrepancy in group numbers was due to exclusions on the basis of virus expression. In these studies, targeting iSPNs with the approach described resulted in a greater number of exclusions because the GPe is bordering the DMS and so any expression of retro-Cre within the DMS, or mCherry expression in the GPe necessitated exclusion. We have clarified this in the subsection “Analyses”. The reviewer’s other point that this means that the experiment provided more power for detecting an effect in Group dSPN+CNO is correct; however, the important point to make here is that it was specifically in the dSPN+CNO group that we failed to find a significant devaluation effect (i.e. a significant difference between valued vs. devalued). The fact that we found a deficit in this group despite having more power while, conversely, finding a significant devaluation effect in the iSPN group despite having less power, indicates that differences in power do not explain the observed effects.

- In Figure 4, they map unilateral manipulations of the striatal pathways to responding on the right or left side of the chamber. However, while the levers are on different sides of the chamber, the animals are freely moving and can press either lever however they want, with either contralateral or ipsilateral limbs or directional actions. So, despite the lever position in the chamber, it is not clear if the lever-pressing is in fact ipsilateral or contralateral to the stimulation side. The effect also appears to be driven most strongly by a side bias in the dSPN eNPHR group when the stimulation is off.

This is a good point. The Reviewer is correct that the animals were freely moving and could press the lever however they wanted, and we acknowledge that we did not (could not with our current set-up) look at whether each animal used their ipsilateral or contralateral limb or a combination of both to press the levers; we did not have the video resolution to assess this validly. Nevertheless, supporting our conclusions, the results reported here accord (i) with previous reports of lateralised lever pressing with unilateral manipulations (e.g., Tai et al., 2012), (ii) reports of increased dopamine release in the hemisphere contralateral to the lever position in a lever pressing task (Parker et al., 2016; Lee et al., 2019), and (iii) our own demonstration of increased EGR1 in the hemisphere contralateral to the lever (see Figure 4—figure supplement 2). In all of these examples, there were no restrictions on which paw the animals used to press the lever. What these studies and our experiment have in common is the location of the lever relative to the magazine, and this appears to be critical; animals generally move between checking the magazine and pressing the lever which means that, for the contralateral lever, this necessitates a contralateral turn from magazine to lever, which is likely to be the source of the increased bias. Given the importance of this point, we have now added it to our discussion..

- No discussion is made on potential motor or rotational effects of stimulating these pathways in their optogenetic experiments, which was quite strong (10mW) and high frequency (40Hz). Changes in performance may be due to non-specific rotational behavior disrupting the animal's ability to perform these tasks, which was not measured or discussed.

To address this point, recordings of all test sessions were scored for rotational behaviours. We saw no evidence of LED-induced rotation in any animals. While we acknowledge that rotational effects have been described for light delivery in the dorsal striatum, these have only been observed using continuous light delivery (7mW and 15mW light delivery), and not for pulsed light delivery (15mW light pulsed at 20Hz) (Owen et al., 2017). We have included an analysis of rotational behaviour taken from the test session videos in the results. See Figure 4—figure supplement 2 I and J. See also Results.

- There were some scholarship issues with citations being incorrectly attributed. For instance, the manuscript incorrectly cites Kim et al., 2009 with the sentence "bilateral stimulation of dSPNs tends to increase, and of iSPNs to retard, both spontaneous movement and the flexible performance of reward-related actions". There was no stimulation or cell type specific experiments in this paper. In this same sentence they incorrectly cite Hikida et al., 2010, which did not use stimulation but reversible synaptic transmission blockade of each pathway, and Kwak and Jung, 2019, which also used inhibition and did not report on motor actions of stimulating these pathways. I didn't exhaustively check every cited reference but these jumped out at me as I knew the papers well. The authors should check that all references are correctly attributed.

We apologise for this oversight. The sentence in the introduction to which the reviewer refers was left unmodified from an earlier draft of the manuscript. We have now modified the sentence for accuracy to read:

“These studies have found that bilateral manipulation of dSPNs and iSPNs in the pDMS produces either differential effects on spontaneous movement or on the flexible performance of reward-related actions (Kravitz et al., 2010; 2012; Nonomura *et al*., 2018; Hikida et al., 2010; Kwak and Jung, 2019), whereas unilateral dSPN and iSPN stimulation favors movements and the selection of actions in contralateral and ipsilateral action space, respectively (Kravitz et al., 2010; Bay Konig et al., 2019; Tai et al., 2012).”

Reviewer #3:<break />In this study, the authors examined the role of the dorsomedial striatum in the encoding and updating of goal-directed behaviour. Using a set of circuit specific complementary approaches, the authors found that learning of goal-directed behaviour recruits the medium size spiny neurons of the direct pathway (dSPN) whereas updating of this learning specifically recruits the medium size spiny neurons of the indirect pathway (iSPN). The topic investigated by the authors is important. The methods are appropriate and elegant and the experiments very well executed and analysed. I think that the results significantly add to the field. My comments are listed below:<break />1) The authors included both male and female animals. Did they find any indication of a gender effect?

This is an interesting question. We found no clear effects of gender throughout our main training and test data – see Author response table 2. Within each group, we compared male and female subjects, focusing on the key measures at each stage of training or testing. In only two groups were there gender-related differences; in Group iSPN+CNO there was a significant difference in overall press rate (female > male) during instrumental training and in Group CNO control there was a significant difference in magnitude of devaluation (female > male) on the choice extinction test. These groups did not show gender differences on any other test and so, when combined with the fact that other groups did not show gender differences on these specific tests, we believe that there was no substantial evidence for an effect of gender and so have not included gender-related comparisons in the manuscript.

**Author response table 2. resptable2:** 

Experiment	Training/Test stage	Group	Stats
DREADDs suppression	Instrumental training (press rate; male vs female)	CNO control	F(1,11)=2.43, *p*=0.147
		dSPN+VEH	F(1,14)=0.27, *p*=0.616
		iSPN + VEH	F(1,2)=4.613, *p*=0.165
		dSPN+CNO	F(1,10)=0.15, *p*=0.710
		iSPN+CNO	**F(1,4)=11.55, *p*=0.027**
	Non-rewarded choice test (magnitude of devaluation; male vs female)	CNO control	**F(1,11)=5.425, *p*=0.040**
		dSPN+VEH	F(1,14)=1.41, *p*=0.254
		iSPN + VEH	F(1,2)=1.02, *p=*0.419
		dSPN+CNO	F(1,10)=0.17, *p*=0.685
		iSPN+CNO	F(1,4)=3.97, *p*=0.117
DREADDs stimulation	Instrumental training (press rate; male vs female)	mCherry+CNO	F(1,4)=0.41, *p*=0.556
		hM3D+VEH	F(1,4)=0.07, *p*=0.806
		hM3D+CNO	F(1,6)=0.60, *p=0.815*
	Non-rewarded choice test (magnitude of devaluation; male vs female)	mCherry+CNO	F(1,4)=2.44, *p=0.913*
		hM3D+VEH	F(1,4)=0.30, *p*=0.612
		hM3D+CNO	F(1,6)=0.64, *p*=0.454
Optogenetic inhibition	Instrumental training (press rate; male vs female)	eYFP control	F(1,9)=0.00, *p*=0.975
		dSPN eNpHR	F(1,4)=4.98, *p*=0.090
		iSPN eNpHR	F(1,6)=0.43, *p*=0.534
	Non-rewarded choice test (sex x LED interaction on magnitude of devaluation)	eYFP control	F(1,9)=0.03, *p*=864
		dSPN eNpHR	F(1,4)=0.00, *p*=1.00
		iSPN eNpHR	F(1,6)=0.13, *p*=0.734
	Choice test with unilateral stimulation (gender x LED effect on lever bias)	eYFP control	F(1,9)=0.197, *p*=0.668
		dSPN eNpHR	F(1,4)=0.067, *p*=0.809
		iSPN eNpHR	F(1,6)=0.170, *p*=0.694
	Reversal (difference between rewarded and non-rewarded; male vs female)	eYFP control	F(1,9)=0.01, *p*=0.931
		dSPN eNpHR	F(1,4)=0.28, *p*=0.628
		iSPN eNpHR	F(1,6)=0.63, *p*=0.458

2) Past research has demonstrated that the anterior and posterior portions of the DMS are differentially involved in the control of goal directed learning and performance. As a result, the figures showing virus infections should include the anatomical boundaries of the infections along the antero-posterior axis.

We have now included schematics showing the extent of virus spread across the antero-posterior axis of the dorsal striatum for all DREADDs experiments (see Figure 2—figure supplement 1E and Figure 3—figure supplement 1B). As evident in these schematics, virus expression is primarily evident in the pDMS (as described in Hart and Balleine, 2018). We acknowledge that there is expression that spreads into what might be considered the more posterior regions of aDMS; an unavoidable consequence when injecting a substantial volume of virus in order to fill a large structure such as the DMS mediolaterally. It is worth noting, however, that despite some anterior spread, there was little evidence of spread beyond +1.00 mm anterior of bregma. For optogenetic experiments, we have mapped virus expression in the pDMS only, which extends posteriorly from +0.24mm anterior to bregma (Hart and Balleine, 2018), as optogenetic inhibition was also dependent on the location of the cannula placement; animals for which visual confirmation of cannula placement was outside the pDMS were excluded from behavioural analyses. Having said that, we didn’t set out in these studies to directly compare posterior with anterior DMS and so we don’t think it would be appropriate to draw further conclusions here.

3) The design of Experiment 2 (see Figure 2) with A1-O3, A2-O3 followed by A1-O1, A2-O2 might create some sort of outcome identity reversal learning. Could the authors comment on the absence of iSPN manipulation?

The reviewer raises an interesting point. We agree that this training produces a change in outcome identity across sessions although it doesn’t produce a *reversal* in identity. As such there is no conflict between pretraining and subsequent training. What evidence we have to date suggests that it is the conflict induced by identity reversal that is most effective in creating interference with new encoding. For example, iSPN manipulations impair the updating of goal-directed learning when the identity of two A-O contingencies is reversed (Matamales et al., 2020) and, in the current study, when the reward contingencies are reversed (see Figure 5). When goal-directed learning depends on encoding two novel A-O associations and there is no identity conflict, it appears that is not sufficient to impair new learning. We now touch on this point in the discussion in a combined paragraph addressing this and point 4 below. See the final paragraph of the discussion prior to the conclusions:

4) Previous research of the same group has provided evidence for a role of cholinergic striatal interneurons (CIN) in the updating of goal-directed behaviour. Could the authors include a discussion about the possible relative contribution of CIN and iSPN neurons in this process?

This is a good suggestion by the reviewer and we have added a short paragraph to the discussion addressing this point together with point 3 above. There is indeed evidence that, like iSPNs, cholinergic interneurons in the pDMS play a role in the updating of goal-directed learning; disconnection of the parafascicular thalamus to CINs in the pDMS impairs rats’ ability to update goal-directed learning when the identity of two action-outcome associations are reversed (Bradfield et al., 2013). As is noted in that study, any effect of altered CIN function must become manifest through changes in the plasticity of spiny projection neurons. Critically, thalamostriatal connections generate burst-pause firing in CINs, which has a profound effect on the cortical regulation of SPNs. Following a brief period of transient presynaptic inhibition of both dSPNs and iSPNs, there follows a period of enhanced postsynaptic excitability in iSPNs but not dSPNs driven by the activation of M1 muscarinic receptors (Ding et al., 2010). Thus, thalamic activation of CINs in the pDMS creates a temporal window in which the striatal network is biased towards the cortical activation of neighbouring iSPNs only and a loss of function of either appears to abolish the capacity to update goal-directed learning. See the Discussion.